# CMIP7 Data Request: Impacts and Adaptation Priorities and Opportunities

Alex C. Ruane[1], Charlotte L. Pascoe[2], Claas Teichmann[3], David J. Brayshaw[4], Carlo Buontempo[5], Ibrahima Diouf [6,] Jesus Fernandez[7], Paula L.M. Gonzalez[8], Birgit Hassler[9]. Vanessa Hernaman[10], Ulas Im[11], Doroteaciro Iovino[12], Martin Juckes[13], Iréne L. Lake[14], Timothy Lam[15], Xiaomao Lin[16], Jiafu Mao[17], Negin Nazarian[18,19,20,] Sylvie Parey[21], Indrani Roy[22], Wan-Ling Tseng[23], Briony Turner[24], Andrew Wiebe[25], Lei Zhao[26], Damaris Zurell[27]

[1] *NASA Goddard Institute for Space Studies, New York, 10709 USA*
[2] *UKRI Science and Technology Facilities Council, Harwell, Didcot, OX11 0QX, UK*
[3] *Climate Service Center Germany (GERICS), Helmholtz-Zentrum Hereon, 20095 Hamburg, Germany*
[4] *Department of Meteorology, University of Reading, Reading, RG6 6ET, United Kingdom*
[5] *European Centre for Medium-Range Weather Forecast (ECMWF), Bonn, 53175, Germany*
[6] *University Cheikh Anta Diop (UCAD), Senegal and University of Labé (UL), Guinea[7] Instituto de Física de Cantabria (IFCA), CSIC-Universidad de Cantabria, 39005 Santander, Spain*
[8] *Met Office, Exeter, EX1 3PB, United Kingdom*
[9] *Deutsches Zentrum für Luft- und Raumfahrt (DLR), Institut für Physik der Atmosphäre, Oberpfaffenhofen, Germany*
[10] *Climate Intelligence, Commonwealth Scientific and Industrial Research Organisation, Aspendale, Victoria 3195, Australia*
[11] *Aarhus University, Department of Environmental Science/iClimate, Roskilde, 4000, Denmark*
[12] *Foundation Euro-Mediterranean Center on Climate Change, CMCC, Bologna, 40131, Italy*
[13] *Kellogg College, University of Oxford, Oxford, OX2 6PN, UKRI STFC, Harwell Campus, Didcot, OX11 0QX, UK and National Centre for Atmospheric Science, Leeds, UK*
[14] *Swedish Meteorological and Hydrological Institution, SMHI 601 76 Norrköping, Sweden*
[15] *Potsdam Institute for Climate Impact Research (PIK), 14473 Potsdam, Germany*
[16] *Kansas Climate Center, Kansas State University, Manhattan, 66506, Kansas, USA*
[17] *Environmental Sciences Division, Oak Ridge National Laboratory, Oak Ridge, TN 37830, USA*
[18] *School of Built Environment, University of New South Wales, Sydney, NSW 2052, Australia*
[19] *Australian Research Council Centre of Excellence for Climate Extremes, NSW 2052 Australia*
[20] *Australian Research Council Centre of Excellence for the 21st Century Weather, NSW 2052 Australia*
[21] *Electricité de France, EDF Lab Saclay, 91120 Palaiseau, France*
[22] *University College London, Department of Earth Sciences, London,WC1E 6BT, UK*
[23] *Ocean Center, National Taiwan University, 10617 Taipei, Taiwan*
[24] *CMIP International Project Office, European Space Agency, Harwell, Didcot OX11 OFD*
[25] *Riskthinking.AI, Toronto M6G 2K9 Canada*
[26] *Department of Civil and Environmental Engineering, The Grainger College of Engineering, University of Illinois Urbana-Champaign, Urbana, Illinois 61801 USA*
[27] *University of Potsdam, Institute of Biochemistry and Biology, 14469 Potsdam, Germany*

*Submit to Geoscientific Model Development - Special Issue for CMIP7*
*Original Submission July 15th, 2025*
*Revision October 7th, 2025*
*Updated for Production Files November 3rd, 2025*

*Correspondence to*: Alex C. Ruane (alexander.c.ruane@nasa.gov)

**Abstract.** The Coupled Model Intercomparison Project Phase 7 (CMIP7) undertook an extensive process to gather community input and refine data requests related to impacts and adaptation applications of Earth System Model (ESM) outputs. The Impacts and Adaptation (I&A) Data Request Team worked with CMIP7 leadership to distribute an open solicitation across many communities that use climate model outputs requesting inputs for new and existing variables, the most applicable temporal characteristics, and groupings of variables that together allow for specific application opportunities. This input was then collated and translated into CMIP7 standard templates for inclusion in the broader data request, leading to 13 I&A data request opportunities, 60 variable groups and 539 unique variables sought by vulnerability, impacts, adaptation, and climate services user communities. Here, we describe these opportunities and variable groups, as well as new insights into how ESM groups can prioritize outputs that set off a chain of further analyses, ultimately informing decisions impacting society and natural systems. These include an emphasis on high-resolution outputs to allow further modeling of climate impacts at regional and local scales, improved representation of extreme weather events, enhanced accuracy of downscaling and bias-adjustment techniques, and support for more detailed assessments for decision-making in adaptation and mitigation strategies. There is also broad interest in more extensive provisioning of two-dimensional variables at the Earth's surface, prioritizing experiments that enhance our understanding of both the recent past and future scenarios, and providing outputs that allow further downscaling and bias adjustment. We emphasize that variable groups are the fundamental level at which to engage with the I&A data request, matching the scale of input and the way output provision enables specific I&A applications. Given resource constraints, we applaud CMIP7 efforts to foster strong engagement and communication between ESM groups and the I&A team to build consensus around prudent compromises in priority variables, temporal resolutions, simulation experiments, time subsets, and ensemble members.

# 1 Introduction

## 1.1 Background

Recent climate extremes and record-breaking global temperatures emphasize that climate change is a present and urgent challenge to nature and nearly every aspect of society (Calvin et al., 2023). Mitigation and adaptation actions require clear-eyed insight into the ways that humans are altering the Earth's climate and modifying specific conditions that affect the systems we care about (Ruane et al., 2022). Climate change information must also be timely to enable proactive planning, implementation and system transitions.

The Coupled Model Intercomparison Project (CMIP) has served as the premier protocol-based community for understanding Earth System Models (ESMs) and coordinating ensemble-driven climate assessments. In its previous Phase 6 (Eyring et al., 2016), CMIP6 data requests emphasized the importance of producing data that would be useful for vulnerability, impacts, adaptation and climate services (VIACS) communities (Ruane et al., 2016). This reflected that a major motivation for building and running ESMs is to understand how climate changes affect natural and human systems (Eyring et al., 2019). Inputs from a number of impact sectors and climate services experts in the lead up to CMIP6 pointed to a set of prioritized

climate variables (including several variables new since CMIP5) anticipated to be of primary interest to many communities and a number of variable groups tailored to the needs of select communities. In practice, there was a disconnect between the climate variables available from CMIP6 and the needs from various Impact & Adaptation (I&A) communities (Craig et al., 2022). This disconnect can be attributed to several factors: the mismatch between the spatial and temporal resolutions of CMIP6 outputs and the requirements of impact assessments, which often demand higher-resolution data to inform local adaptation strategies. Additionally, the lack of standardized data formats, bias correction and downscaling techniques, pre-computed ensembles and metadata has posed challenges for users attempting to integrate CMIP6 data into their models and decision-making processes. Addressing these gaps is crucial for enhancing the usability of climate model outputs in real-world applications.

CMIP7 (Dunne et al., 2024) features new and improved ESMs that take advantage of more powerful computational and data storage capabilities. Within the CMIP7 framework, data request processes have been conducted to prepare efficient and effective variable production for 35 registered Model Intercomparison Projects (MIPs) and other applications within the VIACS communities (which also extends well beyond CMIP7). The CMIP7 design recognizes that VIACS communities are rising in prominence given their critical role in climate action planning and implementation, and they are growing in expertise, organization, and engagement with both stakeholders and the physical climate science community.

**1.2 ESMs in applications context**

In practice, CMIP's relevance to impacts and adaptation planning is vital but not always direct. ESM outputs provide the fundamental climate response information that launches a long sequence of processing and analysis that supports risk management and adaptation planning (Figure 1) (Doblas-Reyes et al., 2021). This process is initiated by scenario inputs (e.g., emissions and land use information associated with socioeconomic and geopolitical pathways) provided by groups such as the integrated assessment models associated with ScenarioMIP (Van Vuuren et al., 2025). These outputs are then converted into greenhouse gas, aerosol, and land use forcing fields that will serve as driving conditions (as overseen by the CMIP7 Forcings Task team and input4MIPs). The ESMs then simulate the responses of the Earth system to these scenarios, generating foundational understanding of changes to the atmosphere, ocean, biosphere, cryosphere and land surface potentially including all of the variables requested by this CMIP7 I&A team. ESMs rely on some of the world's most powerful computing systems, yet they must remain efficient in order to simulate long time periods, large ensembles, multiple models, different model versions (with varying structures and physics), and a wide range of future scenarios. The importance of ESM efficiency can lead to coarse outputs that motivate further downscaling (Gutowski et al., 2016; Jones et al., 2024; Yu et al., 2022) and bias-adjustment (Karger et al., 2017; Lange, 2019; Thrasher et al., 2022) to capture important sub-grid-scale responses and reach decision relevant temporal and spatial scales, although these can also introduce delays into assessment planning. This information is often then passed into impacts models that represent the response of human and natural systems (e.g., agriculture, water resources, human health, infrastructure, energy, fisheries, ecosystems), further translating global change information into the units of asset risks and benefits that may govern stakeholder decisions. Outputs from all these models are

then interpreted, translated, and communicated by experts, boundary organizations, climate services and practitioners, allowing
for the design and implementation of climate responses (Doblas-Reyes et al., 2021). Climate services communities therefore
depend on ESM outputs and help connect stakeholders with the climate information they need to inform risk management,
adaptation, and mitigation on a variety of time horizons (e.g., World Meteorological Organization (WMO) Climate Services,
NOAA, UK Met Office, the World Bank Climate Change Knowledge Portal, Copernicus Climate Services).
The flow of climate information from ESMs to adaptation-relevant decision-making is complex, non-linear, and
increasingly flexible. Figure 1 illustrates that the processing pipeline is not a fully closed system; information can enter or exit
at various stages. Many applications are designed to bypass certain steps to conserve resources or address methodological
limitations. For instance, some workflows move directly from ESM outputs to bias-adjusted datasets without intermediate
dyanmical downscaling. Other worksflows may utilize emulators as simplified, computationally efficient models
approximating the behavior of full ESMs using statistical or machine learning methods trained on existing ESM outputs. While
they cannot  replace ESMs entirely, they allow users to explore additional scenarios  or sensitivity analyses with reduced
computational cost. While these shortcuts may introduce simplifications or degrade signal quality, emerging technologies—
such as artificial intelligence and physics-informed machine learning—offer promising pathways to bridge gaps efficiently,
provided they are applied with care (Molina et al., 2023; Kashinath et al., 2021).
The ESM step is critical to the entire processing and decision-making pipeline, motivating great interest in the CMIP7
variable request from many downstream communities. Lack of information from ESMs cuts off whole areas of downstream
application, so it is critical that CMIP7 establish an early, close and sustained engagement with the VIACS communities. Even
where emulators may be used to skip the ESM step, these would depend on foundational ESM simulations with robust outputs
to train the emulators across all requested variables. Machine learning and other data-driven approaches are only as good as
the underlying datasets, which underscores the importance of CMIP's original, physics-based information about climate
response. The steps following the ESM simulations also take substantial time and effort, which emphasizes the need for timely
production and sharing of CMIP outputs, particularly in light of pressing (though not yet finalized at time of writing) IPCC
AR7 deadlines.

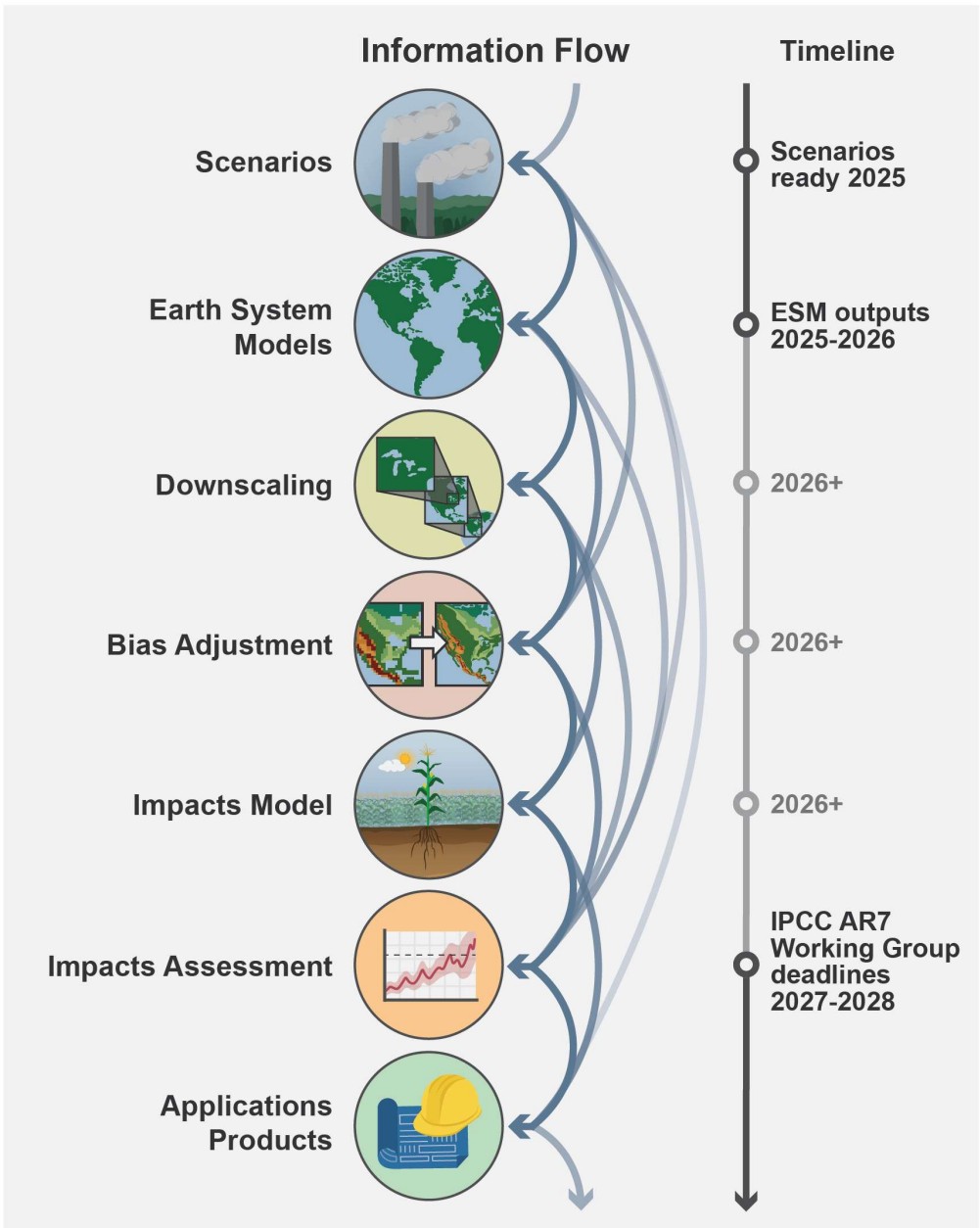


**Figure 1**: ESM outputs in the broader context of information flow and analysis for impacts and adaptation applications. Dark arrows indicate a direct chain of inputs and outputs for analyses developing scenarios into applications products, while lighter arrows indicate the potential for skipped steps utilizing methods such as artificial intelligence or emulators to accelerate the process and conserve resources. The timeline on the right indicates targeted milestones that puts time pressure on the larger process (although precise dates are not yet finalized by the IPCC and UNFCCC), with gray dates indicating reduced certainty given dependence on resources and the successful completion of previous steps and accessibility of outputs. Ruane and Kozlowski (2025).


## 2 Approach and methodology


The I&A Theme is unique within the CMIP7 Data Request because of its need to reach communities far beyond those
developing and running ESMs. Work thus began by recruiting a team that could speak to the experience and practice of CMIP
applications for nature and society. CMIP7 began by drawing from the foundation of the VIACS Advisory Board initiated in
CMIP6 (Ruane et al., 2016), opening further with an open solicitation for participation that attracted dozens of applicants. The
author team was selected based on expertise in the I&A application of CMIP outputs, an interest in covering major application
sectors (e.g., water resources, cities, agriculture, ecosystems, infrastructure, health), community-led projects, and regions
(Bernier et al., 2024). The team also selected      authors who could utilize established networks to reach the broader VIACS
communities to maximize input and community representation in the CMIP data request process.

The I&A Team was especially interested in soliciting feedback from non-traditional CMIP partners and users of ESM
outputs; however, this led to a larger gap in terms of familiarity with CMIP interfaces and data tools. Rather than asking the
broader VIACS community experts to navigate the Climate and Forecasting (CF) Standard forms and CMIP7 AirTables, which
define variables, variable groups and opportunities, the I&A team worked with the CMIP International Program Office (IPO)
to establish a Mural Board web link for community input. Mural Board operates like a white board for generating priority data
requests, with space for respondents to leave notes indicating variables of interest, provide a short justification of prioritization,
and to describe the community/perspective and application they were representing. This process allowed aggregating input
into relatively broad opportunities, each covering inputs from several community expert  groups, many of which are not (yet)
as well organized as the ESM community. The Mural Board was divided into boards for 11 sectors (Climate Services,
Agriculture, Cities, Fisheries, Human Health, Infrastructure, Marine Fisheries, Terrestrial and Freshwater Ecosystems, Peace
and Socioeconomic Development, Water Resources, Energy) and a Miscellaneous/Other board. These Mural boards provided
instructions, examples, and templates to make I&A data requests as easy and accessible as possible, while also collecting
contact information associated with each entry to document reach and allow for follow-ups when clarification was necessary.
The I&A Author team then sent out invitations to provide input across broad author and CMIP networks (**Figure 2**). Outreach
followed a multi-pronged strategy to advertise the effort and solicit community input, including personal conversations,
workshop and conference announcements and sessions, direct emails, messages on project and community list-servs, social
media alerts, communications and special events organized by the CMIP IPO, and bulletins on WCRP websites (e.g.,
https://www.wcrp-climate.org/). In many cases, these advertisements were further disseminated through partner networks
reaching thousands of impacts and adaptation experts (e.g., via CORDEX, AgMIP, ISIMIP). The I&A Author team also filled
in data request needs determined by their own experiences, conversations with colleagues in their field, and information drawn
from  groups that have published data requirements in the literature (e.g., for the storm surge community; Bernier et al., 2024).
The co-development effort was also aided by web announcements, emails, and direct invitations from the CMIP IPO.
The overall I&A outreach response successfully included direct inputs on each sectoral Mural Board, with
professional and scientific communities being the most common respondents, likely reflecting internal deliberations within
groups already organized for collaboration in a climate applications realm. Where gaps were identified the team conducted
more direct follow-on outreach, although it is likely that groups with additional inputs did not explicitly engage in the process.
As inputs were public at the time of solicitation, some later inputs built on or clarified earlier inputs. The I&A Team tracked
this iterative process to help determine shared priorities when combining variable requests.

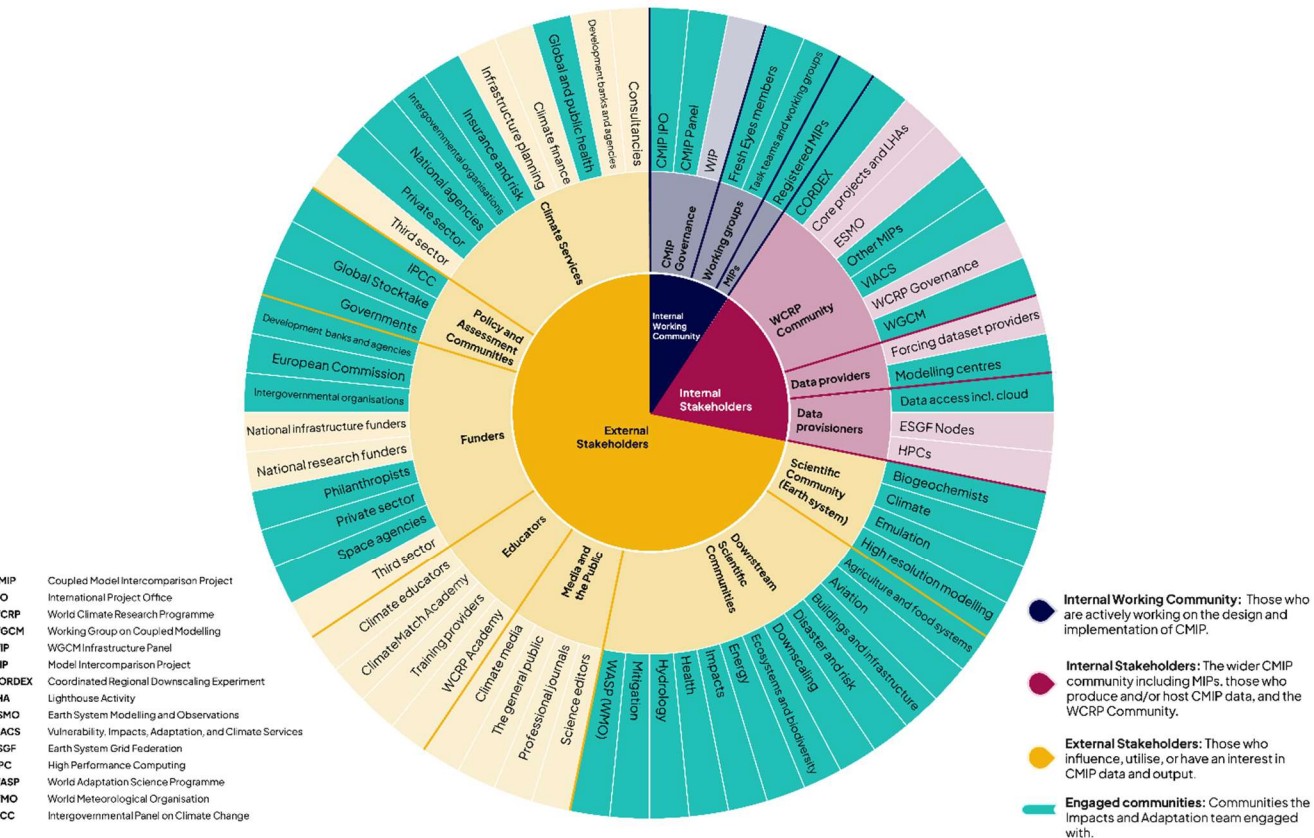

**Figure 2**: Categories of communities contacted and contributing to the I&A Data Request, with additional informal interactions furthering
the reach of the input process. The bottom-right legend indicates the broad categories of stakeholders as colored in the innermost circle,
with those categories elaborated outward and the final ring indicating which communities the I&A Team specifically engaged. Ruane et al.
(2025)

## 3 Information management and decision making

Inputs from the climate applications community allowed the I&A Team to summarize the VIACS perspective to engage ESM groups in planning output production. These contributions came both from individuals responding directly to the solicitation and from communities that had deliberated internally before submitting collective feedback. The vast majority of Mural Board inputs were akin to variable groups, providing a list of specific variables required to perform a given application, for example capturing the breadth of climatic impact-drivers needed to assess risks for a vulnerable asset, calculating an impacts-relevant metric, or driving an impacts model. In some cases, the same Mural Board input included several groupings of variables for distinct applications or indications of broad support for existing requests.

Harmonizing the I&A data request with the broader CMIP7 Data Request efforts required translating the Mural Board inputs into the CF Standard forms for proposing new variables and the CMIP7 AirTables for proposing variable groups and opportunities (Mackallah et al., 2025). This process began with an initial review of inputs and communication with community members to clarify any potentially confusing entries and to track down any missing information. The I&A team then merged similar variable groups according to the type of application that each group made possible. Finally, variable groups were clustered into coherent opportunities that generally aligned with the major impacts sectors. Reflecting the collaborative nature of the process, opportunities were named according to topic rather than for specific communities, projects, or models. Multiple opportunities were constructed in broad sectors like infrastructure, which had distinct output requirements for aviation and ground transportation applications. I&A opportunities were then scrutinized by independent CMIP7 data request authors, resulting in an additional round of consolidation to avoid overwhelming ESM groups deciding on their output targets (Appendix A). The details of the other thematic areas and variable groups included in the Data Request are provided in the companion manuscripts in Atmosphere (Dingley et al., 2025), Land and Land Ice (Li et al., 2025), Earth System (McPartland, 2025), and Ocean and Sea Ice (Fox-Kemper 2025) themes.

## 4 Results: Content included in the CMIP7 Data Request

### 4.1 I&A Opportunities and variable groups

The I&A Data Request process produced 13 I&A opportunities (Table 1) consisting of 60 variable groups.

**Table 1**: Overview of all opportunities associated with the Impacts and Adaptation data request (further details on variable groups provided in opportunity motivations below).

| Section | ID | Opportunity title | #Variable Groups | Opportunity Description |
|---------|-----|-------------------|------------------|-------------------------|
| 4.1.1 | 19 | Agriculture and Food System Impacts | 10 | Supports agricultural sector simulations and analyses of extreme event response, air pollution, water and land use for I&A planning in agriculture and food systems. |
| 4.1.2 | 77 | Bias-adjustment for impacts modelling and analysis | 4 | Enables bias reduction procedures that convert ESM outputs to more relevant scales for broad use by I&A applications. |

| 4.1.3 | 16 | Climate impacts on marine biodiversity and ecosystems | 4 | Supports marine biodiversity and ecosystem expert efforts to assess climate change impacts in marine ecosystems and for different ecological groups. |
|---|---|---|---|---|
| 4.1.4 | 20 | Core climate services | 2 | Baseline high-frequency output to support diverse communities working in climate services and climate impact modeling as a vital step in the value chain for understanding and managing the risks of a changing climate to society. |
| 4.1.5 | 81 | Dynamical Downscaling | 7 | High-resolution climate simulations performed with regional climate models (RCMs) and convection-permitting regional climate models (CPRCMs) are more and more used to feed regional climate services on a wealth of applications at fine spatial scales. |
| 4.1.6 | 80 | Empirical Statistical Downscaling and Emulators | 4 | Machine learning-based statistical approaches are now used to emulate the behavior of high-resolution climate models such as RCMs or CPRCMs. When trained, those tools have the capacity to replace the dynamical downscaling approaches for a variety of applications such as creating multi-member ensembles or emulating an emissions scenario not run with the emulated RCMs. |
| 4.1.7 | 22 | Energy system impacts | 7 | Supports energy sector simulations relating to, e.g., system operation, system planning, renewable power resources, use and availability of hydropower, transmission and storage planning, energy usage patterns, climate resilience of generation assets (including nuclear), and adaptation to extreme events. |
| 4.1.8 | 37 | Health impacts | 6 | Allows for further analysis of human health impacts across a range of major hazard types, including heat and humidity extremes, flooding, and air pollution. |
| 4.1.9 | 41 | Impacts of climate change on aviation | 1 | Supports studies focused on the impacts of climate change on the aviation sector. Tracks climate changes to upper tropospheric winds and temperature that govern jet streams and, in turn, affect wind shear and clear air turbulence as shown in multiple studies. |
| 4.1.10 | 42 | Impacts of climate change on transport infrastructure | 1 | Supports studies focused on the impacts of climate change on the transport sector (railways and roads, mainly). The sector is severely impacted by extreme weather events, through multiple hazards such as heat, flooding, extreme winds, coastal erosion and storm surges. Future climate projections are used to inform business planning, adaptation and regulatory submissions by the transport infrastructure sector. |
| 4.1.11 | 11 | Terrestrial Biodiversity | 4 | Supports model-based impact assessments and adaptation planning in the terrestrial biodiversity and ecosystem services sector. This spans a broad set of impact models from global to regional scales. |
| 4.1.12 | 40 | Vulnerability of urban systems, infrastructure and population | 2 | Enables studies focused on the impacts of climate variability and change on urban infrastructure and population. With more than half the global population living in cities and this number being projected to rapidly increase, understanding the projected changes in hazards to urban environments such as heat and flooding is of vital importance to support effective adaptation strategies. |
| 4.1.13 | 67 | Water security and Freshwater Ecosystem Services | 16 | Supports future projections of hydroclimatic hazards and their effects on freshwater ecosystems and water resources to plan adaptation actions and resilience building measures that ensure water security, ecosystem services and planetary health. |


Many of the I&A opportunities cover a large number of applications appealing to broad audiences for use of CMIP7 outputs.
Opportunities often align with IPCC Working Group II sectoral chapter topics, which in turn have a large number of sub-topics
with unique variable groups and potential applications that could stand alone as their own opportunities. The I&A Data Request
Team emphasizes that ESM groups should engage with the Opportunities first to determine the types of applications they
would hope their model outputs would support, and then select among Variable Groups according to their own judgment
balancing resource constraints, scientific interests, and motivation to support particular impacts and adaptation decision
processes.  As the I&A Team expects decisions operating on the Variable Group level, some variables are included in multiple
variable groups to ensure internal coherence and sufficiency for the intended application (see also Section 5.1). Redundancies
of output variables across variable groups are seen as highlighting a benefit given that it indicates multiple uses for the same
output and a reduced overall burden for the ESM teams. The I&A Team also encourages ESM groups to avoid thinking that
opportunities are all-or-nothing; even providing variables for one Variable Group (or a subset of those variables) is valuable.

In the following sections we describe the motivations for each I&A Data Request Opportunity and variable group,

deepening the summaries and justifications provided in the CMIP7 Data Request AirTable.

### 4.1.1 Opportunity ID 19: Agriculture and Food Systems Impacts

Variables in this Opportunity support applications that help understand and prepare for risks to regional and global food
systems, food security, and the livelihoods of many rural populations that rely on agricultural systems (Mbow et al., 2017;
Pörtner et al., 2022). Agricultural systems, here focused on terrestrial practices including field crops, agroforestry, and animal
agriculture, have great potential for adaptation and mitigation as the agricultural sector is a major emitter of greenhouse gases
and driver of land cover change. Forewarning of future risks allows for proactive planning and the development of adaptation
and mitigation strategies that will be effective, timely, and just, even when responses are non-linear or influenced by non-
climatic factors (Ruane et al., 2024; Zhao et al., 2024)

Large communities of practice are well-established in using climate model outputs for agricultural applications. This

requested received multiple inputs from the Agricultural Model Intercomparison and Improvement Project (AgMIP), which is
an independent (non-CMIP7) community undertaking multi-crop-model ensemble projections of future agricultural yields,
water requirements, and the downstream ramifications on food prices, food security, socioeconomic development, and
geopolitical stability (Rosenzweig et al., 2013; Ruane et al., 2017; Hasegawa et al., 2018). AgMIP also coordinates global
gridded crop model simulations (Jägermeyr et al., 2024) in collaboration with the Inter-Sectoral Impacts Model
Intercomparison Project (ISIMIP; Frieler et al., 2024). Additional applications utilize empirical or machine learning models to
anticipate crop risks or viability under different environments (Sweet et al., 2025).

The opportunity includes 10 variable groups (Table 2), beginning with an AgModelCoreDaily variable group that

includes the most commonly required information to drive a process-based crop model. This core set also serves as a minimal
dataset for a large number of additional agricultural indices and suitability analyses. The AgModelExpandedDaily variable
group allows for more complex determination of evapotranspiration and drought conditions in agricultural lands. An
AgModelHourly variable group adds sub-daily information for more complex agricultural modeling efforts (see Appendix B
for more information on new variables from all I&A opportunities). The opportunity also features variable groups that explore
specific aspects of agricultural systems in a changing world, including a focus on carbon stocks and fluxes, agricultural water
resources (core and extended), air pollution impacts, and land use change. The AgTile variable group requests tile water and
energy balance information corresponding to the agricultural components of a larger grid cell, and the AgImpactsMisc variable
group includes additional variables related to wind gusts and monthly $CO_2$ concentrations.

**Table 2**: Motivation for variable groups associated with Agriculture and Food Systems Impacts

**ID 19: Agriculture and Food System Impacts**
**Brief description:** Supports agricultural sector simulations and analyses of extreme event response, air pollution, water and land use for impacts and adaptation planning in agriculture and food systems.

| Variable group | Reason for inclusion |
|---|---|
| AgModelCoreDaily | Core daily variables needed to run majority of processed-based crop models such as those included within the Agricultural Model Intercomparison and Improvement Project (Rosenzweig et al., 2013; Ruane et al., 2017). |
| AgModelExpandedDaily | Additional variables that allow more complex agricultural modeling, including enhanced representation of radiation balances and evapotranspiration. |
| AgCarbon | Variables needed to assess carbon stocks and greenhouse gas exchanges on agricultural lands including crops and pastures. |
| AgWaterCore | Outputs that allow for evaluation of demand for water resources in agricultural lands. |
| AgWaterExt | Additional variables that allow more detailed evaluation of soil water balances and demand for water resources. |
| AgModelHourly | Sub-daily variables that are needed for more complex agricultural models and analyses, allowing for more precise resolution of diurnal cycles and agroclimatic extremes. |
| AgAirPollution | Conditions for aerosols and ozone that can damage agricultural systems. |
| AgTile | Indications of sub-grid-scale temperature, water, and energy balances for agricultural tiles that may be substantially distinct from the broader grid condition. |
| AgImpactsMisc | Surface wind and $CO_2$ concentration information requested by agricultural community. |
| AgLandUse | Variables describing land use classifications for agricultural lands |


**4.1.2 Opportunity ID 77: Bias-adjustment for impacts modeling and analysis**
This opportunity requests variables that are critical to the pipeline of bias-adjustment for impacts model assessment (Figure
1). Variable groups were consolidated from community feedback from groups including ISIMIP (Frieler et al., 2024) and
variables for the Climatologies at High-resolution for the Earth's Land Surface (CHELSA) project (Karger et al., 2016, 2017).
ISIMIP produced bias-adjusted climate datasets from CMIP6 outputs (Lange, 2019) for coordinated use by impacts models
across numerous sectors (e.g., agriculture, human health, forests, fisheries, water resources, energy).   The requested
ISIMIPMinimalAtmos variable group will enable the CMIP7 generation of ISIMIP climate projections allowing for analysis
of multi-sectoral risk. ISIMIP also requested the ISIMIP3hourlyAtmos variable group given interest by regional models that
operate at finer temporal resolution. The ISIMIP_deposition_variables group includes wet and dry deposition variables (NH3,
NH4, NO$_y$) useful for ecosystems, agriculture, and water resources analyses. CHELSA utilizes mechanistic statistical
downscaling to produce fine resolution (1km) mean climate statistics that have proven useful for analysis of biodiversity,
wildlife and ecosystem services that may be highly sensitive to niche climate conditions in areas with complex topography,
heterogeneous land use or coastlines. The CHELSA_land_daily provides outputs that would continue this work with CMIP7
outputs.

Variable groups included in this opportunity were developed with strong input from developers of ISIMIP and

CHELSA (Table 3). Their established applications communities, many of whom have developed their own processing pipeline
to convert climate projections into impacts model input datasets, make further application of CMIP7 outputs likely. These
outputs will also enable other existing products (e.g., NASA NEX; Thrasher et al., 2024) and new bias-adjustment and
statistical downscaling approaches that may emerge in the coming years, potentially including novel machine learning
methods.

**Table 3**: Motivation for variable groups associated with Bias-adjustment for impacts modeling and analysis.

| ID 77: Bias-adjustment for impacts modeling and analysis | |
|---|---|
| **Brief description:** Enables bias reduction procedures that convert ESM outputs to more relevant scales for broad use by impacts and adaptation applications. | |
| **Variable group** | **Reason for inclusion** |
| ISIMIPMinimalAtmos | Variables requested by the Inter-Sectoral Impacts Model Intercomparison Project (ISIMIP; Frieler et al., 2024) and likely to be of use for other downscaling methodologies. This minimal set is sufficient to drive bias adjustment for the bulk of impacts models across the ISIMIP sectors. |
| ISIMIP3hourlyAtmos | Sub-daily variables requested by the Inter-Sectoral Impacts Model Intercomparison Project (ISIMIP) in order to run subset of more complex impacts sector models. |
| ISIMIP_deposition_variables | These variables are used to track atmospheric $NH_3$, $NH_4$, and $NO_y$ wet and dry deposition which can affect ecosystems, agriculture and water systems. |
| CHELSA_land_daily | Minimal set of variables needed as input for the CHELSA downscaling procedure (Karger et al., 2016), which is widely utilized by many ecosystems impact assessments and also feeds into part of the ISIMIP protocol. |


### 4.1.3 Opportunity ID 16: Climate impacts on marine biodiversity and ecosystems
This opportunity provides outputs of interest to marine biodiversity and ecosystem modellers assessing climate change impacts
on marine ecosystems for different ecological groups (Cooley et al., 2022). The variables and experiments in this opportunity
provide input to a  broad set of fisheries and marine ecosystem models within the Fisheries and Marine Ecosystem Model
Intercomparison Project (FishMIP; Tittensor et al., 2018, 2021) and beyond. Mariculture is a growing industry expected to
further expand as the need for protein increases globally. Fisheries models, which focus on both wild populations, ecosystems
and mariculture, play a crucial role in fisheries management, conservation, and the understanding of marine ecosystems (Doney
et al., 2012). For example, these models can evaluate the impact of temperature variability on fishery sustainability (Wang et
al., 2020), helping to evaluate habitat suitability and population management strategies, ensuring sustainable practices. This
opportunity provides critical data to enhance the models and assess future environmental shifts and corresponding effects on
marine life. Daily resolution and extended variables can significantly enhance the accuracy of fish abundance estimates.
Factors such as algal primary production, dissolved oxygen and the depth of the 17ºC isotherm can provide a more
comprehensive understanding of fish distributions. By integrating these variables, the model can offer more insight into aquatic
life population responses to changing climate.

The opportunity thus supports a broad set of use cases to model marine ecosystem services (Table 4). The variable
group Marine_bgc_baseline is the baseline list of biogeochemical variables needed to fully characterise the marine ecosystem.
The variable group biodiv_marine_daily is requested to provide inputs for marine biodiversity models. The variable group
ISIMIP_oceanforcing_3hr is required for bias-adjustment of the oceanic forcings provided within ISIMIP. To date, oceanic
forcing is typically not bias-adjusted although that is quite critical for the regional marine ecosystem and fisheries models that
are calibrated by observational data (Frieler et al., 2024; Lengaigne et al., 2024).
**Table 4:** Motivation for variable groups associated with Climate impacts on marine biodiversity and ecosystems.

| ID 16: Climate impacts on marine biodiversity and ecosystems | |
|---|---|
| **Brief description:** Supports marine biodiversity and ecosystem expert efforts to assess climate change impacts in marine ecosystems and for different ecological groups. | |
| **Variable group** | **Reason for inclusion** |
| Biodiv_marine_daily | Daily climate variables for biodiversity and ecosystem modeling in marine realm |
| ISIMIP_oceanforcing_3hr | The 3-hourly atmosphere-ocean surface fluxes (freshwater, heat, and momentum) will allow for a bias-adjustment of the oceanic forcings provided within ISIMIP. |
| Marine_bgc_baseline | This is the baseline list of variables that is needed to fully characterize the marine ecosystem. Each one plays an important role in validation, monitoring, comparison against observations, and understanding ecosystem services. |
| Marine_bgc_fishmip | Collection of data to support the modeling of marine ecosystem and fisheries to investigate changes and impacts |


### 4.1.4 Opportunity ID 20: Core Climate Services

Kennedy et al. (2024) noted that the "delivery and use of climate information to enable climate action has never been more
crucial". Activities within climate service and climate impact analyses tackle a diverse range of challenges, spanning different
sectors, scales, and actors. It is conducted both in public and private organizations, and may take the form of relatively general
user interface platforms or more bespoke activities for specific use cases. This field is rapidly growing - in 2024, at least 55
National Meteorological and Hydrometeorological Services provided climate services across 19 sectors (up from 45 in 2020;
Kennedy et al., 2024). Provision of a base set of high-quality near-surface climate variables is thus expected to be taken up
by a very large and diverse community to improve the understanding and assessment of climate risk and impacts on key sectors.
The need for high-quality climate information cuts across many areas from water and agriculture to energy, financial
services and health. Climate services are vital steps in this chain, supporting the assessment, understanding and interpretation
of climate change in the context of specific sectoral applications. The impacts considered are often highly complex and require
fine-resolution modeling in both time and space. However, because of the nature of climate impacts work, many different
applications depend on a relatively small set of common surface/near-surface meteorological variables – for example, health
risks from heatwaves and the demand for energy are both strongly sensitive to near-surface temperatures and humidity (Taylor
and Buizza, 2003; Simpson et al 2023). In previous CMIP rounds, the availability of sub-daily input data relating to these
fields has been patchy, inhibiting impact modeling for climate services by requiring compromises on the temporal resolution
of impact modeling activities or the extent of inter-model comparison (e.g., restricting analysis to only the subset of CMIP
models providing output at the required granularity). In more extreme cases, lack of available data may even prevent the use
of CMIP simulations in high-quality impact models completely (e.g., the absence of suitable high-frequency data has stymied
some energy-system applications; Craig et al., 2022). This opportunity seeks to provide high-quality, high-frequency
outputs to underpin a wide range of climate impact modeling activities to support enhanced climate service provision.
The Opportunity includes 2 variable groups (Table 5), both including high-frequency data, with both also requested
in Opportunity 22 (Energy System Impacts) and enabling additional applications. The first group,
impact_climserv_hourly_core focuses on a small set of sub-daily near-surface properties - temperature, wind-vectors, wind
gusts, precipitation, humidity, insolation - which are suitable to support diverse applications in, e.g., health, energy, water,
insurance. The second group, impact_climserv_hourly_expanded extends this to support a broader range of applications or
more detailed impact models (e.g., building energy use, hydrological applications, splitting solar power between concentrating
and PV types). Hourly frequency is indicated, but this is intended to be interpreted as a request for the "best possible" sub-
daily resolution. Note that variable properties are selected according to the use case need (e.g., instantaneous sampling vs
time-averaging, spatial sampling) and preserving these aspects is important for physical consistency and intercomparison.
Experiments are selected to have most relevance to real-world challenges within the energy sector. Significant time subsets
(with a minimum 20-30 years) and ensemble sizes (as many realizations as possible) are requested to ensure statistical
robustness of analysis (e.g., to separate model uncertainty and internal variability). It is noted throughout that single-precision
output is very likely sufficient for most impact applications.

**Table 5**: Motivation for variable groups associated with Core Climate Services.

| ID 20: Core Climate Services | |
| --- | --- |
| **Brief description:** Climate services and climate impact modeling are a vital step in the value chain for understanding and managing the risks of a changing climate. This outlines a minimum variable set of high frequency outputs to support detailed process-based impact models to revolutionize climate projection data for service provision. | |
| **Variable group** | **Reason for inclusion** |
| impacts_climserv_hourly_core | Core sub-daily (ideally hourly or best available frequency) output to support onward modeling with sophisticated impact models across a range of sectoral applications. |
| impacts_climserv_hourly_expanded | Additional sub-daily (ideally hourly or best available frequency) output to support onward modeling with sophisticated impact models across a range of sectoral applications. |


### 4.1.5 Opportunity ID 81: Dynamical Downscaling

High-resolution climate simulations using regional climate models (RCMs) are increasingly vital for informing climate
services across a range of societal and environmental applications. Dynamical downscaling will play a crucial role in bridging
CMIP7 global projections and the needs of the impacts, adaptation, and vulnerability communities. The 3D, model-level data
requested from the historical and scenario experiments are essential to perform dynamical downscaling. Among the key
initiatives providing such data, the Coordinated Regional Climate Downscaling Experiment (CORDEX), endorsed by the
WCRP, is recognized for its standardized approach, delivering authoritative downscaled outputs over 14 nearly continental-
scale domains, derived from ESM simulations (Gutowski Jr et al., 2016). A global effort under CORDEX-CMIP6 has already
produced over 250 simulations (with 600+ planned, https://wcrp-cordex.github.io/simulation-status), underlining the
widespread demand for standardized boundary forcings.
Dynamical downscaling by means of RCMs (Giorgi, 2019), the core methodology in CORDEX, relies on high-
frequency (typically 6-hourly) three-dimensional boundary conditions from ESMs, spanning both historical and future scenario
periods. To better capture the range of global warming levels (GWLs), particularly the pre-industrial baseline (GWL+0), an
extension of the temporal coverage historical ESM outputs would be needed. Specifically, we recommend extending the
provision of lateral boundary conditions back to 1850 and forward to 2100 using scenario experiments; then at least up to 2125
using extension experiments. For standard CORDEX domains (10-25 km resolution), 6-hourly ESM data suffice. However,
higher-frequency outputs (e.g., 3-hourly) may be required for future km-scale nesting applications anticipated under
CORDEX-CMIP7, particularly from higher-resolution ESMs (e.g., <40 km grid spacing) as envisioned in HighResMIP
(Roberts et al., 2025).

Apart from the CORDEX output, this opportunity also merges requests from individual downscaling groups including

those interested in regional ocean downscaling. The request has been divided into 7 variable groups (Table 6). The
dynamical_downscaling_core group includes essential 6-hourly 3D atmospheric variables required for general downscaling.
To address the growing demand for refined marine climate projections, the dynamical_downscaling_ocean_core group
provides variables to force regional ocean models, including physics, biogeochemistry, and sea ice. The
dynamical_downscaling_aerosols_chemistry group expands on traditional atmospheric variables by including aerosols and
chemistry-related data, to be used by models including chemical components. For biogeochemical modeling,
dynamical_downscaling_ocean_biogeochemistry_forcing                                                             and
dynamical_downscaling_ocean_air_sea_biogeochemistry_forcing request monthly and daily 3D-global forcing fields,
respectively. Meanwhile, dynamical_downscaling_ocean_air_sea_forcing considers hourly variables at the air-sea interface,
critical for ocean-only model forcing. Finally, dynamical_downscaling_soil addresses the land component by requiring high-
frequency 3D soil temperature and moisture fields.

Given the larger size of 3-dimensional, high temporal resolution variables included in the Dynamical Downscaling

opportunity, we reiterate that ESM groups          can focus on the core variable groups and are encouraged to provide as
many of the additional groups as they deem practical        .    Note that variables provided for      RCM simulations also
enable    further downscaling     by CPRCMs to achieve     higher resolution for specific regions.

**Table 6**: Motivation for variable groups associated with Dynamical Downscaling

| ID 81: Dynamical Downscaling | |
|---|---|
| **Brief description:** Dynamical downscaling activity for CORDEX (and other RCMs and later CPRCMs) requires high-frequency lateral boundary conditions from ESMs for the historical and scenario periods. | |
| **Variable group** | **Reason for inclusion** |
| dynamical_downscaling_core | Downscaling is a key step in filling the gap between CMIP7 global climate model experiments and the activity of the communities dealing with vulnerability, impacts and adaptation studies. Dynamical downscaling requires 6-hourly 3D-global forcing fields for specific variables. This core variable group is the minimum requirement to perform dynamical climate downscaling. |
| dynamical_downscaling_ocean_core | There is an emerging need for regional downscaling of global climate models for the ocean (physics, biogeochemistry, sea ice) to provide refined and more regional-to-local information on marine environment past and projections changes, to support ocean climate services and for climate adaptation.  This is the minimal set of variables needed to force regional ocean models at their lateral boundaries, and at the air-sea interface for forced models. |
| dynamical_downscaling_aerosols_chemistry | ESM outputs needed to drive regional climate models with enhanced atmospheric chemistry. Up to now, the variables needed were mainly limited to atmospheric variables (hus, ta, ua, va, ps) every 6 hours (6hrPlev) as well as some surface variables |

| | (tos, siconc...). This set includes variables associated with new aerosol and chemistry components of the climate system now included in more and more regional models. |
|---|---|
| dynamical_downscaling_ocean_air_se a_biogeochemistry_forcing | Dynamical downscaling of the ocean air-sea biogeochemistry requires daily 3D-global forcing fields for specific variables. |
| dynamical_downscaling_ocean_air_se a_forcing | Dynamical downscaling of the ocean requires hourly forcing fields at the air-sea interface for specific variables. |
| dynamical_downscaling_ocean_bioge ochemistry_forcing | Dynamical downscaling of the ocean biogeochemistry requires monthly 3D-global forcing fields for associated variables. |
| dynamical_downscaling_soil | Includes variables required to drive a coupled land surface model (LSM) that implicitly solves the heat and mass exchanges between the atmosphere and the land surface, including those involving vegetation, which are central to the hydrological cycle. The use of LSMs in regional climate simulations, in turn, requires high frequency 3D fields of soil temperature and moisture at specified soil depths, which ideally should be sourced from the same global climate simulations used to force the atmospheric model. |


### 4.1.6 Opportunity ID 80: Empirical Statistical Downscaling and Emulators

Empirical-statistical downscaling (ESD; Maraun et al., 2019) exploits the relationship between large-scale and local climate
observations to convert coarse ESM output into regional-to-local climate projections. Mathematical models can also be trained
on complex ESM and RCM outputs to reproduce key features at lower computational cost, thus statistically emulating its
behaviour (Baño-Medina et al., 2024). Both ESD and RCM-emulators are increasingly important components of the CORDEX
framework, enabling an efficient generation of high-resolution climate information tailored to regional and local needs. As in
the previous case, this is an intermediate opportunity that is not sector-specific but enables a key step in the information pipeline
described in Figure 1. These CORDEX techniques and similar ESD efforts can serve critical sectors such as agriculture, water
management, energy, and urban planning, where timely, location-specific climate data is vital for decision-making.
Stakeholders include national meteorological services, environmental agencies, infrastructure planners, and adaptation
practitioners. Established communities, like the CORDEX-ESD group or the recently established CORDEX Task Force on
Machine Learning, help coordinate efforts and improve methodological robustness.
This request targets key variables needed for both traditional ESD methods and newer hybrid emulators. (Table 7)
These tools are trained on large-scale atmospheric fields from reanalysis products or driving ESMs and can be applied to
CMIP7 outputs, provided compatible variables are available. The statistical_downscaling_core_daily group includes essential
daily variables (e.g., temperature, wind, moisture, geopotential height) on standardized pressure levels, supporting the main
workflow of empirical-statistical methods that downscale historical and scenario simulations. In response to increasing user
demand for high-frequency climate information, the statistical_downscaling_core_6hourly group covers the same set of
variables at 6-hour resolution. These are crucial for applications such as hydrological modeling, sub-daily event analysis, and
early warning systems. Maintaining consistency in pressure levels across both daily and sub-daily datasets ensures
compatibility in training and deployment phases. The statistical_downscaling_extended group includes variables like monthly
wet-day frequency and mean wet-day precipitation, which, while derivable from daily data, are computationally intensive to
calculate. Their inclusion directly supports efficient modeling of 24-hour precipitation statistics and broader adoption of ESD
by users with limited processing capacity. Finally, the global_emulators group provides variables for training AI-based
emulators that replicate the behaviour of complex climate models. These emulators enable cost-effective, flexible projections,
including under scenarios not originally simulated. AI-driven tools are also expected to play a transformative role in CMIP7
analysis by enabling rapid synthesis across multi-model ensembles.

**Table 7**: Motivation for variable groups associated with Empirical Statistical Downscaling and Emulators (CORDEX).

| ID 80: Empirical Statistical Downscaling and Emulators | |
| --- | --- |
| **Brief description:** This opportunity covers empirical-statistical downscaling (ESD) activities in CORDEX, including hybrid downscaling (emulators). | |
| **Variable group** | **Reason for inclusion** |
| statistical_downscaling_core_daily | Core daily variables are used as large-scale fields for empirical statistical downscaling of climate projections (historical and scenario runs) which are typically at daily timescale and at multiple pressure levels. |
| statistical_downscaling_core_6hourly | Core 6-hourly variables are used as large-scale fields for empirical statistical downscaling of climate projections (historical and scenario runs) which are required for the increasingly demanded sub-daily timescale. |
| statistical_downscaling_extended | Additional variables requested for the convenience of users. They could otherwise be calculated by downloading several 3D fields the variables included in the high priority request. However, the latter would require downloading several 3d-fields. |
| Global_emulators | The requested variables will facilitate the creation of a family of foundation models to enable analysis of CMIP7 simulations using new Artificial Intelligence (AI) methods. |


### 4.1.7 Opportunity ID 22: Energy System Impacts

The de-carbonization of the global energy system is central to climate mitigation, with investment in clean energy reaching
US$2 trillion in 2024 (Giroud, 2024; Hassan et al., 2024, IEA, 2024). The scale of change is fundamentally transforming the
nature of how energy systems operate. The CMIP outputs requested in this opportunity support a wide range of energy-sector
applications seeking to understand, assess and manage climate risk within this complex and rapidly-evolving energy landscape,
"overcoming the disconnect" that has previously existed between the energy- and climate- system science communities (Craig
et al., 2022). Specific applications include the understanding and assessment of system operation, system planning, renewable
power resources, use and availability of hydropower, transmission and storage planning, energy usage patterns, climate
resilience of generation assets (including nuclear), and adaptation to extreme events.
There is an emerging international community of practice to support the uptake of energy systems-relevant ESM
output, including its onward "conversion" to provide vital climate services to end-users and decision-making stakeholders
(WMO, 2022). While no single, definitive WCRP-based "-MIP" exists (perhaps reflecting the aforementioned historic
disconnect between energy and climate scientists; Craig et al., 2022), numerous overlapping groups draw in a wide
community of research, industry and policy-makers. These include the Next Generation Challenges in Energy and Climate
Modelling workshop community (NextGenEC; Craig et al., 2022), the International Conference on Energy and Meteorology
(ICEM), the Open Energy Modelling Initiative (OpenMod), multiple long-running dedicated sessions at the American
Geophysical Union and European Geophysical Union, established provision climate services and datasets (e.g, PECDv4,
Dubus et al. 2022), and relevant parts of ISIMIP.

The Opportunity includes 7 variable groups, sharing a set of high-frequency variable groups with Opportunity 20 (Core Climate Services): impacts_climserv_hourly_core and impacts_climserv_hourly_extended (Table 8). These emphasize a small set of sub-daily output of near-surface fields pertaining to energy applications — ideally hourly but intended to request the "best possible" sub-daily resolution. The core set impacts_climserv_hourly_core provides the baseline information for detailed process-based energy modeling at the whole-system level (i.e., informing the integration of renewable generation and whole-system design), while impacts_climserv_hourly_expanded permits additional modeling refinements, making it possible to distinguish between types of solar power generation and providing additional information for wind power and energy-use modeling. It is noted that the need for high-frequency output is dictated by both scientific necessity and convention. Scientific necessity tracks that power systems require quasi-instantaneous matching of supply, demand and storage across a spatially extended transmission network, thus recommending complex data which maintains "realistic" multi-variate spatio-temporal co-variability that is essential even as statistical downscaling is usually not possible. Convention favors data for detailed energy-system models typically run at hourly resolution, which facilitates greater uptake by the energy modeling community. Variable properties are selected carefully according to need: e.g., instantaneous sampling of wind vectors is essential to avoid smoothing associated with time-averaging that produces substantial biases in wind-power modeling applications. Sub-daily outputs (at least 3-hourly) are also required for: impact_river_temperature (risk to environment and/or curtailment of cooling systems for nuclear and fossil-fuel generation) or impact_energy_building_demand (use of energy in buildings), while daily outputs, including daily maxima, can be used for impact_energy_hydropower (hydropower availability); impact_energy_damaging_wind (wind storm damage to energy infrastructure); impact_energy_damaging_other (proxies for damage to energy infrastructure from other meteorological hazards). Experiments are selected to have most relevance to real-world challenges within the energy sector. Significant time subsets (min 20-30 years) and ensemble sizes (as many as possible) are requests to ensure statistical robustness of analysis (the sensitivity of energy system planning to climate variability on multi-annual timescales has been well established; Bloomfield et al 2016, Zeyringer et al 2018). It is noted throughout that single-precision output is very likely sufficient for most energy applications.

**Table 8**: Motivation for variable groups associated with Energy Systems Impacts.

| ID 22: Energy System Impacts | |
|---|---|
| **Brief description:** De-carbonizing energy systems is a central part of climate change mitigation. However, many of the changes needed - e.g. the shift to renewables - increase the exposure of the energy system to weather and climate. This opportunity supports detailed onward modeling for impacts and adaptation planning in the energy sector. | |
| **Variable group** | **Reason for inclusion** |
| impacts_climserv_hourly_core | Baseline sub-daily (ideally hourly, else best available) information for detailed process-based energy modeling at the whole-system level, allowing evaluation of wind and solar integration, energy use, system operation and whole-system design. |
| impacts_climserv_hourly_expanded | Enhanced sub-daily (ideally hourly, else best available) information for detailed process-based energy modeling at the whole-system level. These variables permit additional modeling refinements over core list, for example distinguishing between types of solar |

| | power generation, and providing additional information for wind power and energy-use modeling. |
|---|---|
| impact_energy_hydropower | Additional daily variables to permit basic modeling of hydropower availability. Many energy systems around the world are heavily dominated by hydropower for electricity production (storage and run-of-river). |
| impact_energy_damaging_other | Additional monthly summary variables to quantify aggregate impact of damaging weather on energy system infrastructure (except wind). |
| impacts_energy_damaging_wind | Additional monthly summary variables to quantify aggregate impact of damaging wind storms on energy system infrastructure. |
| impact_energy_building_demand | Additional sub-daily variables to support enhanced modeling of building demand (including heat pumps). Electricity demand in buildings is expected to increase, both to support mitigation and to sustain the probably unavoidable increase in summertime cooling demand. |
| impact_energy_river_temperature | Additional sub-daily information needed to drive hydrological thermal models that can simulate how electricity generation through thermal power plants use a river for cooling. Excess river temperature leads to generator curtailment (unavailability) or environmental damage. |

452

### 4.1.8 Opportunity ID 37: Health impacts

This opportunity facilitates applications of ESM outputs to gauge impacts and potential adaptation to climatic impact-drivers connected to human health (Ranasinghe et al., 2021) across 6 variable groups (Table 9). Climate change has both direct and indirect impacts on human health. Direct health impacts arise from a number of climatic impact-drivers, including pervasive mean conditions and extreme heat, cold, wind storms, flooding, and ice events, as well as changes to atmospheric composition which influences the air pollution levels humans are exposed to (Im et al., 2022, 2023). Indirect health risks arise from climate's influence on the many natural and human systems that are linked to broader health systems, including effects on vector-borne diseases, food and water safety and security, malnutrition, and the consequences of malnutrition and displacement. The nature of these climate impacts heightens the interests of the general population and a large number of public, private, civil society and non-governmental organizations charged with reactive and proactive planning to reduce morbidity and mortality across complex health systems and diverse, vulnerable populations.

Three variable groups allow for deepening analysis of the effects of extreme heat and cold risks for human populations including heat stroke, heat exhaustion, hyperthermia and hypothermia. First, a core dataset of daily variables (HealthHeatCoreDaily) includes both surface daily maximum temperature and relative humidity information given that many impacts are strongly connected with compound heat and humidity hazards that can challenge the body's ability to self-regulate temperature through perspiration (Staiger et al., 2019). These outputs are also useful for outdoor recreation and tourism applications. Daily resolution is essential, as heat waves are poorly represented by monthly outputs. In addition, minimum temperature data support the evaluation of cold-season hazards such as hypothermia and the calculation of wind chill indices (Kinney et al., 2015). Second, the HeatHealthExpandedDaily variable group is designed to meet the needs of health experts who have developed a large number of heat-health indices to understand risks to different demographic groups and populations exposed through outdoor labor (e.g., agricultural or construction work) (Vanos et al., 2020). This includes further information

on the daily range of humidity, short- and longwave radiation and wind speeds that can drive extremes in body heat or
dangerous wind chills. Third, the HeatIndices variable group requests that daily and maximum daily wet bulb globe
temperature (Knutson and Ploshay, 2016) and NOAA Heat Index (Lin et al., 2012) be calculated within the model so that there
is no methodological confusion in downstream impacts and adaptation applications.
Two variable groups in this opportunity focus on health risks associated with air pollution, which depends strongly
on the type of air pollution and the duration and intensity of exposure (Orru et al., 2017). HealthPollutionDaily captures
information related to mixing ratios for surface particulate matter at 1, 2.5, and 10 micron diameters, surface $NO_2$, and surface
ozone on a daily scale. That set also includes temperature and humidity outputs as these govern the environment for chemical
processes in the atmosphere that may convert precursors into dangerous air pollution constituents. Corresponding hourly
outputs are also requested when possible in the HealthPollutionhourly variable group in order to gauge acute extremes and
interactions with the diurnal cycle of temperature, humidity and precipitation. This information cannot be determined only
from emissions data and projections.
HealthFloodingDaily requests variables important for developing engineering solutions to reduce risks to humans
from inland flood waters. This includes details on maximum hourly precipitation rates, snow water equivalent, soil moisture,
and runoff to describe the pre-conditioning and extent of flooding events. Additional variables relevant to indirect health CIDs
are available in other I&A opportunities related to food systems (ID 19), water systems (ID 67), ecosystems (IDs 11 and 16),
urban populations (ID 40), and broader climate services (ID 20).

**Table 9**: Motivation for variable groups associated with Health impacts.

| ID 37: Health impacts | |
|---|---|
| **Brief description: A**llows for further analysis of human health impacts across a range of major hazard types, including heat and humidity extremes, flooding, and air pollution. Additional variables relevant for health systems are requested in opportunity IDs 11, 16, 19, 20, 40 and 67. | |
| **Variable group** | **Reason for inclusion** |
| HealthHeatCoreDaily | Variables in this group are very commonly used to understand the effects of extreme heat and cold on human health, including compounding events with high humidity. |
| HeatHealthExpandedDaily | Variables in this group allow for more in-depth analysis of the health impacts of heat and cold. Additional variables allow for more complex heat-health analyses (including humidity and radiation) and investigations of wind chill. |
| HealthPollutionDaily | Variables in this group allow health impacts and adaptation experts to track air pollution that are known to be hazardous to human health. Including these variables allows experts to track interactions between the climate system and pollutants that would not be available simply from emissions datasets. |
| HealthPollutionhourly | Impacts modelers are interested in hourly outputs to understand acute air pollution risks to human health. |
| HealthFloodingDaily | Variables in this group provide information about precipitation, soil moisture, snowpack, and runoff that can lead to dangerous flood conditions that can impact human health. |
| HeatIndices | Directly calculated heat indices that facilitate health risk management and adaptation planning for exposed populations. |


**4.1.9 Opportunity ID 41: Impacts of climate change on aviation**
This opportunity was designed to support studies focused on the impacts of climate change on air travel and the broader
aviation sector. Climate change causes changes on the jet streams that in turn affect wind shear and can cause clear air
turbulence (Gratton et al., 2022; Lee et al., 2019; Williams and Joshi, 2013). The variables in this group allow experts to
diagnose clear air turbulence that is of relevance for passenger safety concerns as well as planning of flight routes and times
(Table 10). The 'essential' variable group unlocks the opportunity of diagnosing clear air turbulence from climate model output
from 5 pressure levels (150, 175, 200, 225, and 250 hPa) in the upper troposphere/lower stratosphere (UTLS) region. These
are standard model output variables (horizontal wind components and temperature) but requested at specific vertical levels.

**Table 10**: Motivation for variable groups associated with Impacts of climate change on aviation.

| ID 41:  Impacts of climate change on aviation | |
|---|---|
| **Brief description:** Tracks climate changes to jet streams that in turn affect wind shear and can cause clear air turbulence as shown in multiple studies. | |
| **Variable group** | **Reason for inclusion** |
| aviation_impacts_essential | The variables in this group allow to diagnose clear air turbulence that is of relevance for passenger safety concerns as well as planning of flight routes and times. This can be diagnosed using an essential set of 5 levels in the upper troposphere/lower stratosphere (UTLS) layer: 150, 175, 200, 225, and 250 hPa. |


**4.1.10 Opportunity ID 42: Impacts of climate change on transport infrastructure**
This opportunity provides outputs of interest to the transport infrastructure communities looking to CMIP7 for
projections in order to evaluate impacts, risks, and potential adaptations in the transport sector (railways and road, mainly)
(Table 11). The sector is severely impacted by extreme weather events, through multiple hazards such as heat, flooding,
extreme winds, coastal erosion and storm surges. Future climate projections are used to inform business planning, adaptation
and regulatory submissions by the transport infrastructure sector. This is supported by substantial work in the topic focused on
different regions of the world (de Abreu et al., 2022; UK Climate Change Committee, 2022; Nemry and Demirel, 2012; Palin
et al., 2013, 2021). Climate resilience and efficiency of transport infrastructure also impacts mitigation efforts given
transportation's role in current emissions.

**Table 11**: Motivation for variable groups associated with Impacts of climate change on transport infrastructure.

| ID 42: Impacts of climate change on transport infrastructure | |
|---|---|
| **Brief description:** Provides outputs of interest to the transport infrastructure communities looking to apply CMIP7 projections for risk evaluation, business planning, adaptation, and regulatory submissions by the transport infrastructure sector. | |
| **Variable group** | **Reason for inclusion** |
| Transport_impacts_essential | Requested variables support studies focused on the impacts of climate change on the transport sector (railways and roads, mainly). The sector is severely impacted by extreme weather events, through multiple hazards such as heat, flooding, extreme winds, coastal erosion and storm surges. |


**4.1.11 Opportunity ID 11: Terrestrial Biodiversity**
This opportunity includes variable groups providing information needed for model-based impact assessments and adaptation
planning in the terrestrial biodiversity and ecosystem services sector relevant for initiatives such as the IPCC (Parmesan et al.,
2022) and the Intergovernmental Science-Policy Platform on Biodiversity and Ecosystem Services (IPBES). The included
variable groups will serve a broad range of biodiversity and ecosystem models aiming to understand past biodiversity and
ecosystem changes as well as projecting future trends in biodiversity and ecosystem services. These are applied to evaluate
impacts, risks, and potential adaptations that safeguard biodiversity and ecosystem services and ensure planetary health and
human well-being.

Variable groups in this opportunity provide the minimal set of monthly variables essential for assessing broad-scale

impacts of climate change on biodiversity and ecosystem services as well as advanced sets of variables with daily and sub-
daily resolution essential for understanding long-term dynamics through process-based biodiversity, forest, and biophysical
models (Table 12). Biodiv_land_monthly covers typical inputs needed for coarse-scale biodiversity and ecosystem services
models as summarised in the BES-SIM protocols (Kim et al., 2018; Pereira et al., 2024). Additionally, these variables can be
used to conduct impact assessments for different ecological groups and realms (Hof et al., 2018; Zurell et al., 2018).
Biodiv_land_daily requests variables needed as input for advanced process-explicit and statistical biodiversity models and
process-based forest models (Ferrier et al., 2016). These span models that predict spatiotemporal distribution and abundances
of species or functional types, to models of avian species migration pathways and disease spread; thus supporting a broad set
of use cases for modeling interactions between nature and people. Biodiv_land_daily_advanced provides additional input for
process-based forest models (Grünig et al., 2024). Forests provide crucial ecosystem services to people, help mitigate climate
change impacts, and are affected by synergistic global changes like climate extremes and disease spread. The variable group
supports advanced process-based forest models through provision of variables related to extreme disturbances.
Biodiv_microclim_hourly variables serve as input to biophysical models that help to understand and predict climate change
impacts on species' behaviour, phenology, survival, distribution, and abundance (Briscoe et al. 2023). The variables in this
variable group constitute typical input needed for microclimate and biophysical models. These biophysical models are uniquely
suited to solve climate change biology problems that involve predicting and interpreting responses to climate variability and
extremes, multiple or shifting constraints, and novel abiotic or biotic environments.

**Table 12**: Motivation for variable groups associated with Terrestrial Biodiversity.

| ID 11: Terrestrial Biodiversity | |
|---|---|
| **Brief description:** Supports model-based impact assessments and adaptation planning in the terrestrial biodiversity and ecosystem services sector. This spans a broad set of impact models from global to regional scales. | |
| **Variable group** | **Reason for inclusion** |
| Biodiv_land_monthly | Monthly climate variables essential for biodiversity modeling used in many IPBES analyses. |
| Biodiv_land_daily | Daily climate variables essential for biodiversity modeling, including dynamic vegetation models. The daily resolution is required by several biodiversity models. This variable group also allows the calculation of additional indices relevant for impact assessments on terrestrial ecosystems, including the aridity index, fire weather index, and growing degree days. |

| | |
|---|---|
| Biodiv_land_daily_advanced | Advanced set of daily climate variables for biodiversity modeling, particularly relevant for process-based forest models. |
| Biodiv_microclim_hourly | Minimal set of variables needed as input for physics-based microclimate models essential for ecophysiological models of organisms' heat and water balances. |


**4.1.12 Opportunity ID 40: Vulnerability of urban systems, infrastructure and populations**

This opportunity provides global multi-model output variables that are relevant to urban areas and built environments (Table 13). Cities are both hot spots of climate impacts and fundamental foci of adaptation strategies (Dodman et al., 2022), which has motivated a Special Report on climate change and cities in the IPCC Seventh Assessment Report (AR7) cycle. While only covering 2-3% of the Earth's land surface (Chakraborty et al., 2024; Schneider et al., 2009, 2010), cities accommodate over half of the world's population and contribute to a large proportion of carbon emissions (Creutzig et al., 2015; Seto et al., 2014) - a percentage projected to increase to 68% by 2050 (Birch and Wachter, 2011; Seto et al., 2013). They also respond differently to climate change compared to natural land covers. Due to heat islands, impervious surfaces and lack of arable land, urban environments are even more vulnerable to globally recognized climate risks including heatwaves, extreme precipitation, flooding, water scarcity, and food and energy insecurity (Cao et al., 2016; Georgescu et al., 2021; Li et al., 2024; Zhang et al., 2023; Zhao et al., 2018; Zheng et al., 2021). To address this grand global urban challenge, it is urgent to better understand urbanization and its complex two-way interactions with climate system across spatiotemporal scales. City and infrastructure resilience planning benefits from enhanced understanding of projected changes in physical climate impacts on urban environments, associated biophysical and biogeochemical processes, interconnections with urban microclimates, potential thresholds, and global and regional climate tipping points. Community groups such as the Urban Climate Change Research Network (UCCRN) share best practices in climate output application for impacts and adaptation planning around the world (Rosenzweig et al., 2018; Mahadevia et al., 2025).

The Urban_impacts_essential variable group is the highest priority for urban applications, indicating the minimal set needed to pursue main research and application tasks, with the Urban_impacts_additional variable group enabling deeper analyses of more climatic impact-drivers. Robust characterization of urban climate risks and effective urban planning under climate change relies on robust climate projections specific to built landscapes (Zheng et al., 2021). This opportunity will provide crucial information to robustly underpin those working in urban climate science and impacts assessment, urban governance, urban planning, and sustainable and resilient urban development amidst the challenges posed by a changing climate. This includes improving understanding of city-specific risks in the form of extremes, slow-onset events, and the compounding and cascading of risks in order to inform those working to reduce vulnerability and exposure across urban systems and sectors, including, but not limited to critical infrastructure, buildings, urban ecosystems and biodiversity, food, energy, water and population health. One manifestation of such impacts is the urban heat island (Zhao et al., 2014), which requires detailed sub-daily variables to be modelled. Finally, some requested variable support studies into the urban boundary layer height, which modulates the impacts of air pollution, clouds and precipitation. Furthermore, attribution of these events to rapid urbanization *vs* climate change, as well as the synergies between the two, is key in addressing future risks while

navigating sustainable developments. An increasing number of ESMs and finer-scale models now have an urban representation
and there is interest in tracking how each focuses on processes (ranging from atmospheric turbulence and wind to energy and
water balance) and complexity (ranging from multi-layer parameterizations in the urban canyon to AI/ML), to robustly
characterize uncertainty in global cities.

**Table 13**: Motivation for variable groups associated with Vulnerability of urban systems, infrastructure and populations.

| ID 40: Vulnerability of urban systems, infrastructure and populations | |
|---|---|
| **Brief description:** This opportunity can support studies focused on the impacts of climate variability and change on urban population and the built environment. | |
| **Variable group** | **Reason for inclusion** |
| Urban_impacts_essential | Contains variables needed to analyze impacts to urban environments (both people and infrastructure) e.g. urban heat island, surface flooding, impacts of air pollution. |
| Urban_impacts_additional | Understanding projected feedback from and impacts to urban environments, for example urban heat island, surface flooding, and impacts of air pollution. Additional temporal resolution will enable analyses of onset and development of events. |


**4.1.13 Opportunity ID 67: Water Security and Freshwater Ecosystem Services**
This opportunity identifies variable groups that allow hydrology and ecosystem analysis for water resources, hydropower and
freshwater habitats (Table 14). Climate change increases the risk of floods and droughts and changes groundwater and river
flow regimes with implications for water security, ecosystem services, natural hazards and  human health (Caretta et al., 2022).
Freshwater ecosystems have experienced severe habitat degradation and losses in biodiversity and urgent actions would be
required to halt biodiversity loss and restore ecosystem health following climate-driven changes (Parmesan et al., 2022).

Future projections of hydroclimatic hazards and their effects on the ecosystems and water resources are essential for

assessing what adaptation actions and resilience building measures are needed to ensure water security, ecosystem services
and planetary health. Enhanced modeling, utilizing high resolution climatic data, will aid IPBES in understanding changes in
freshwater biodiversity and ecosystem services and refining scenario development (Ferrier et al., 2016). Large ensemble
simulations are also beneficial in capturing the plausible range of precipitation variability which is crucial for risk assessment
and ensures well-informed adaptation actions (Mankin et al., 2020).

**Table 14**: Motivation for variable groups associated with Water security and freshwater ecosystem services.

| ID 67: Water Security and Freshwater Ecosystem Services | |
|---|---|
| **Brief description:** Supports future projections of hydroclimatic hazards and their effects on freshwater ecosystems and water resources to plan adaptation actions and resilience building measures that ensure water security, ecosystem services and planetary health. | |
| **Variable group** | **Reason for inclusion** |
| WaterResources_daily | Flood and drought risk assessment at daily timescale. Future projections of hydroclimatic hazards at regional scale are essential for assessing adaptation actions and resilience building measures needed to ensure water security. |
| WaterResources_monthly | Flood and drought risk assessment at monthly timescale. The frequency required depends on the region; monthly data might suffice in some places (notably in the tropics where interannual variability is often more pronounced). Monthly data may also appeal to regions and experts with limited computational capacity. |

| WaterResources_subdaily | Flood and drought risk assessment at sub-daily timescale. Some high-impact hazards, such as flash flooding, occur at very short timescales. |
|---|---|
| hydro_modelling_daily | Minimum variable set for hydrological models at daily timescale. Hydrological models simulate the quantity and quality of water in terrestrial water bodies, allowing projections of water security and hazards which will enable assessment of adaptation actions. |
| hydro_modelling_subdaily | Minimum variable set for hydrological models at sub-daily timescale. Hourly precipitation is shown to enable more robust hydrological simulations. |
| impact_energy_hydropower | Many energy systems around the world are heavily dominated by hydropower for electricity production (storage and run-of-river). |
| WaterResourcesPET_daily | Daily variables for calculating potential evapotranspiration (PET). |
| WaterResourcesPET_Monthly | Monthly variables for calculating potential evapotranspiration (PET). |
| biodiv_freshwater_daily | Minimal variable set to allow model-based impact assessments on freshwater biodiversity and ecosystems |


**4.1.14 Other proposed opportunities with Impacts and Adaptation Relevance**

An additional 16 opportunities proposed by other CMIP7 Data Request thematic teams have relevance for impacts and adaptation. In many cases these focus on dynamical structures, surface conditions, or extreme events that are often of great interest in VIACS applications. Cross-thematic evaluation of opportunities identified variable groups that could be combined or merged into opportunities focused on impacts and adaptation. The overall approach continued to recognize that some redundancy in variables across opportunities was acceptable as different variable groups request specific combinations of variables required for proposed analyses. Table 15 summarizes these I&A-adjacent opportunities, with the impacts and adaptation motivation for two opportunities provided to demonstrate their added relevance.

**Table 15**: Overview of all opportunities associated with the Impacts and Adaptation data request (further details on variable groups provided in example opportunity motivations below).

| ID | Opportunity title | #Variable Groups | Impacts and Adaptation Relevance |
|---|---|---|---|
| 24 | Advancing Wind Wave Climate Modelling for Coastal Zone Dynamics, Impacts, and Risk Assessment | 5 | Addresses the need for detailed, accurate modeling and understanding of wind-driven ocean surface waves and how these evolve on global and regional scales, which is essential for predicting coastal hazards, erosion, and other wave-related impacts and supporting offshore renewable energy activities. |
| 38 | Assessments for Hydrological Processes, Water Resources, and Freshwater Systems | 8 | Input data for hydrology, lake, or glacier models, etc. and also variables to assess the usability of Earth system models for water resources applications. Variables also help efforts to evaluate whether these processes/systems are adequately represented in modeling frameworks. |
| 69 | Baseline Climate Variables for Earth System Modelling | 4 | This list reflects the most heavily used elements of the CMIP6 archive, which included many I&A applications. It is intended as a resource for ESM-MIPs developing requests to enable greater consistency among MIPs and as a reference for modeling centers to enhance consistency within MIPs. |
| 44 | Changes in marine biogeochemical cycles and ecosystem processes | 8 | Marine ecosystems are important for the Earth System and absorb around 50% of the atmospheric $CO_2$ in long-term sequestration of carbon. Marine systems are also susceptible to climate change impacts such as heat stress, ocean acidification, habitat loss, and other effects often connected to shifts in biogeochemistry. Relevant for ecosystem process tracking and downstream applications in marine food webs, food production, and tourism. |
| 26 | Detection and Attribution | 3 | Basic variables needed for quantifying changes to mean climate and its variability over time and for understanding the mechanisms involved. Essential for assessing the behavior of models and understanding the role of forcings in the climate system, opening up possibilities to extend detection and attribution into the human and natural impacts realm. |

| 64 | Diagnosing temperature variability and extremes | 2 | Variables requested to more completely diagnose temperature variability, particularly hot extremes, and the processes involved. Facilitates comparisons with observations and intercomparison across models. |
|---|---|---|---|
| 68 | Effects and Feedbacks of Wind-Driven Ocean Surface Waves Coupled Within Earth System Models | 5 | Wind-driven ocean surface waves influence air-sea interactions, coastal hazards, and ocean circulation, affecting climate predictions and extreme event modeling. Accurate representation of waves within Earth System Models improves impact assessments and adaptation strategies for coastal resilience and risk management. |
| 35 | Glacier changes, drivers, and impacts | 5 | Allows more detailed analysis of drivers for sea level rise, water resource provision for freshwater ecosystems, agriculture, and human settlements, as well as risk assessment for glacial lake outburst floods that can devastate downstream communities. |
| 54 | Multi-annual-to-decadal predictability of the Earth System and risk assessment of climate extremes | 4 | Variables requested for experiments that determine predictability within the climate system on different time horizons allow for further investigations of predictability in human and natural systems, particularly when including variables that allow further impacts modeling or calculation of climatic impact-driver indices related to actionable system outcomes. |
| 47 | Ocean Changes, Drivers and Impacts | 20 | Variables help to understand changes in ocean properties and circulations and to evaluate budgets of heat and salt. The Meridional Ocean Circulation (MOC), the Mixed Layer Depth (MKD) and the polar ocean properties are of specific interest and include variables also relevant for ocean ecosystem and fisheries applications. |
| 49 | Ocean extremes | 6 | Outputs to investigate the frequency and severity of extreme ocean conditions driven by climate changes, and support management strategies and adaptation measures to mitigate risks to marine ecosystems and coastal communities |
| 55 | Rapid Evaluation Framework | 1 | Variables requested for timely evaluation and benchmarking of newly available CMIP7 Assessment Fast Track (CMIP7 AFT) simulations as soon as they are uploaded to ESGF, which will help the VIACS communities get early indication of output utility and major findings. First version of the Rapid Evaluation Framework (REF) will be limited to a temporal resolution of monthly mean data and about five metrics/diagnostics per realm. |
| 82 | Robust Risk Assessment of Tipping Points | 2 | Allows for applications related to human and natural system consequences of exceeding tipping points in the climate system, including potential adaptations and adaptation limits. |
| 31 | Role of fire in the Earth system | 2 | Daily and/or hourly air pollutant outputs ($PM_{2.5}$, $O_3$, $NO_2$) to investigate the acute impacts of fires on human exposure and associated health, ecosystem, and agricultural impacts. |
| 63 | Synoptic systems | 2 | Variable groups allowing tracking of synoptic systems can help show how changing dynamical features such as storm tracks, blocking events and atmospheric rivers drive impacts and motivate adaptation planning. |
| 5 | Understanding the role of atmospheric composition for air quality and climate change | 10 | Air pollution levels and composition determine the chronic and acute impacts of air pollution ($PM_{2.5}$, $O_3$, and $NO_2$) on human health, ecosystems and agriculture. These pollutants are also among the short-lived climate forcers. |


*Opportunity ID 55: Rapid Evaluation Framework*

The CMIP Rapid Evaluation Framework (REF) is cross-listed with the Earth System data request (McPartland et al., 2025).

REF was created to evaluate and benchmark the newly available CMIP7 AFT simulations as soon as they are uploaded to the
Earth System Grid Federation (ESGF) with metrics and diagnostics that are available through different open-source evaluation
and benchmarking tools (Hoffman et al. 2025). This opportunity contains the set of variables that are needed for the planned
diagnostics and metrics for the REF (Table 16; CMIP Model Benchmarking Task Team, 2024). The suggested REF
metrics/diagnostics to be available for all CMIP7 AFT experiments are in the first instance very basic evaluations and are not
expected to require very specific variables. The exact selection of variables was also made consistent with the model evaluation
diagnostics in Working Group I Chapter 3 of the latest IPCC report (Eyring et al., 2021). Due to the fixed timeline for the
CMIP7 AFT simulations there is only a short time period for technical implementation of the REF, and therefore the available
metrics and diagnostics in this first version of the REF will be limited to a temporal resolution of monthly mean data and about
five metrics/diagnostics per realm based on a community selection. REF variables and metrics will also support initial
evaluation of formats and fundamental climatologies relevant for VIACS investigations, with broad findings helping to identify
models, time periods, and conditions that merit more extensive investigations when more complete outputs become available.
The realms were chosen specifically to be consistent with the realms used for the data request. Find more information about
the REF opportunity in Dingley et al. (2025).

**Table 16**: Motivation for variable groups associated with the Rapid Evaluation Framework.

| ID 55: Rapid Evaluation Framework | |
| --- | --- |
| **Brief description:** Evaluate and benchmark the newly available CMIP7 AFT simulations as soon as they are uploaded to ESGF. The first version of the REF will be limited to a temporal resolution of monthly mean data and about five metrics/diagnostics per realm. | |
| **Variable group relevant to I&A** | **Reason for inclusion** |
| ref_impacts_and_adaptation | There are 5 total variable groups in this opportunity, but this variable group is the set of variables requested for the planned I&A diagnostics and metrics for the Rapid Evaluation Framework. The variable group will be linked with the "Rapid Evaluation Framework" opportunity, and is essential for the evaluation of the new CMIP7 AFT simulations on a routine basis. |



*Opportunity ID 49: Ocean extremes*
While the CMIP7 Ocean and Sea Ice paper Fox-Kemper et al. (2025) provides a detailed examination of ocean extremes and
their underlying processes, it is important to also recognize them here due to their significant role in impacts and adaptation
studies which are critical for improved understanding of future climate vulnerability and informing robust adaptation strategies.
As the frequency and intensity of extreme ocean conditions - such as marine heatwaves, low-oxygen zones, extreme sea levels,
and storm surges - continue to rise globally, they pose significant threats to marine and coastal ecosystems, coastal
infrastructure and communities, human health, industries and global economies (Bernier et al., 2024; Fox-Kemper et al., 2021;
Gruber et al., 2021; Holbrook et al., 2022). For example, ocean extremes can disrupt fisheries, degrade critical habitats like
coral reefs, kelp forests, and seagrass beds, and intensify coastal flooding and erosion. Studying ocean extremes enables
researchers, managers and policy makers to identify vulnerable regions, assess risks to infrastructure and ecosystems, and
inform policy decisions aimed at building resilience in a rapidly changing climate. The Ocean Extremes opportunity was
developed to request the critical climate model variables at the necessary spatial and temporal resolution that underpins impacts
and adaptation modeling and assessments (Table 17). This, in turn, supports informed decision making, empowers stakeholders
to develop proactive adaptation strategies, build resilience in vulnerable regions facing climate change and better understand
their sensitivity to anthropogenic forcing through a wide range of climate scenarios. This information is critical to assess
impacts of ocean extremes and accurately determine the extent and nature of required adaptation measures. In the coming
decades, ocean extremes and associated floods and erosion are likely to remain a leading cause of natural disasters due to the
effects of the increasing frequency and intensity of extremes combined with increased coastal development associated with
greater exposure (Bernier et al., 2024). .

**Table 17**: Motivation for variable groups associated with Ocean extremes.

| **ID 49: Ocean extremes** | |
|---|---|
| **Brief description:** Supports modeling and assessment of ocean extremes important for identifying vulnerable regions, designing coastal protection to appropriate levels, assessing infrastructure and ecosystem viability, and sensitivity testing adaptation measures. | |
| **Variable group** | **Reason for inclusion** |
| ocean_acidification_oxygen_extremes | Improves understanding of historic and projected changes in pH, oxygen, and salinity as well as associated interactions and impacts. |
| ocean_KE_vorticity_extremes | Enhances understanding of mesoscale eddies and their influence on ocean extremes |
| ocean_Temperature_extremes | Builds understanding of long-term trends and extreme events (marine heatwaves) in terms of historical and projected changes in horizontal (geographic) and vertical (depth layers) occurrence, drivers, frequency and intensity. |
| sea_level_extremes | Tracks ocean extremes, coastal inundation and erosion that can drive vulnerability and adaptation responses in coastal communities and ecosystems. |
| surgemip_variables | Enables analysis of storm surges, coastal inundation and erosion for coastal communities and ecosystems via direct analysis, using downstream/offline tools, or forcing high-resolution regional ocean-wave models, etc. |
| mixed_layer_extremes | Variables with enhanced focus on conditions within the well-mixed upper ocean column that include an abundance of marine life. |


## 4.2 I&A Variables

These I&A opportunities and variable groups included 539 unique varabiles, including 93 new variables defined for CMIP7
(Appendix B). Several variables were not easily added given that they are not yet established in CF-standards, and were thus
left for Data Request v2 (see further discussion in Section 5.4).
Most requested variables are similar to previous CMIP requests, but many new variables were existing physical
parameters now defined at higher temporal resolution. This is motivated by the increasing complexity and precision of many
impacts models, as well as by a strong interest in compound extremes that can result from conditions not resolved at the daily
or monthly resolution. For example, with the growing share of intermittant renewable electricity generation, a precise analysis
of flexibility and storage needs demand that sub-daily consideration of peak loads are balanced with the concomitant available
generation. Therefore, electricity balance models work at a sub-daily resolution and generally use at least hourly observations
or hourly reanalysis products for the current climate. Additionally, building and facilities design relies on regulations which
require the estimation of extreme values, expressed in terms of return levels, and based on the highest values of each variable
under consideration (for wind: extreme wind gust or daily maximum 10-min average wind speed). Information on humidity is
of utmost importance for determining thermal comfort even in indoor environments with heating, ventilation and air conditions
systems, with temperature and air quality also affecting system selection. Since extreme temperature and extreme humidity
may not occur at the same time, detailed sub-daily evolutions of both variables are necessary.
**5 Discussion**
**5.1 Prioritization Process**
Given limited resources (in personnel, funding, computational systems, data storage and time) for ESM groups, the I&A team
organized a set of direct consultations with ESM leaders to communicate VIACS community priorities for output data
production. Operating under the assumption that resources would remain constrained, the I&A Team placed a strong emphasis
on feasibility, striving to balance the need for comprehensive climate information with the practical limitations of data storage,
processing capacity, and model complexity.

**5.1.1 Core and extended variable groups**
Many I&A opportunities include a core/minimum variable group that is of highest priority as determined by variables that (1)
ESMs are likely to be capable of producing, and (2) impacts and adaptation communities will use for nearly all applications in
this opportunity. An ESM contributing the core/minimum variable group will increase the likelihood of that ESM's outputs
informing the broadest applications in that opportunity, while electing to skip these variables may result in that ESM's outputs
being left out of prominent applications in that sector. Opportunities with core/minimum variable groups are accompanied by
variable groups that serve to enhance or extend related applications. ESM groups are therefore encouraged to consider
contributing these deeper variable groups (e.g., more variables, higher temporal resolution outputs) to increase the likelihood
of that ESM's outputs being used in more complex applications and impacts modeling studies, some of which may provide
critical levels of detail that elevate these studies in planning processes. ESM groups may be motivated to include more niche
applications areas that are of particular interest for that ESM development team (e.g., carbon cycles, extreme events,
atmospheric chemistry) or that allow for more tailored impacts and adaptation decision support. In many cases there are
substantial overlaps and only a small number of additional variables are needed to create deeper understanding to enable
concrete applications.

**5.1.2 Temporal resolution**
Many important extreme events happen on time scales finer than one day (e.g., extreme rainfall, heat extremes, acute air
pollution exposure) or have important covariance on a sub-daily scale (e.g., heat and humidity extremes for drought), yet
obtaining model outputs on these time scales is a common stumbling block for VIACS communities. This directly challenges
the use of models and decision support metrics that require sub-daily data (Jägermeyr et al., 2021; Zabel and Poschlod, 2023)
(e.g., crop models, biophysical models, ocean and wave models, river temperature models, models of energy demand in
buildings, electricity system balance models.), as well as applications built around short-interval extreme events (e.g., storm
water flooding that may be governed by maximum hourly rainfall rate). Without this high-resolution information experts have
often employed various types of bias adjustment, turned to downscaling models, or resorted to imposing the climatological
diurnal cycle or stochastic extreme events.  In some cases – for example, certain electricity system planning applications where
coherent sets of multivariate sub-daily time series across an extended geographical domain are essential - the absence of high-
frequency output has been a complete barrier to the use of CMIP data outputs in I&A modeling. In requesting sub-daily outputs,
the I&A Team is requesting the 'best possible' resolution: hourly is better than 3-hourly, but 3-hourly is better than 6-hourly
which is better than daily. As will be discussed in Sect 5.1.4, modeling groups may find efficiency by using time slices on the
highest resolution simulations.
Although some ESM groups may feel that their models were not designed for analysis on fine temporal time steps,
this output would likely be bias-adjusted with a procedure that needs to know unique aspects of change for sub-daily variances,
extremes and covariances that would not be available in daily and monthly outputs. For groups concerned about the data burden
of high-resolution ESM outputs, we note that the I&A high resolution variable requests are mostly 2-dimensional variables
with smaller sizes within the 3-dimensional ESM. We also emphasize that ESMs should not dismiss opportunities given
calculations of overall data sizes, as an opportunity's size may be skewed by its largest variable group while a smaller
core/minimal variable group should also be considered.

**5.1.3 Experimental relevance**
A large number of CMIP7 MIPs are directly relevant to impacts and adaptation (Dunne et al., 2024). While I&A opportunities
and variable groups would be appreciated from any ESM in any of the MIPs, priority is placed on a smaller subset of MIP
experiments. The Historical simulations and 21st century ScenarioMIP simulations (van Vuuren et al., 2025) simulations are
by far the most desired for impacts and adaptation analysis given that they are most directly relevant to the recent past, the
present, and planning procedures covering the coming decades. This information was also conveyed to the CMIP7 panel which
helped motivate the inclusion of ScenarioMIP in the CMIP7 Fast Track which will support rapid analysis ahead of approaching
deadlines for the Intergovernmental Panel on Climate Change Seventh Assessment Report (IPCC AR7)(Figure 1).
Many groups expressed interest in a 100 year outlook (out to 2125), noting that the last several CMIP experiments
have kept 2100 as the distant time horizon even as that horizon gets ever closer. The 2125 time horizon is also important to
understand the ramifications of overshooting in the coming century, which may require a combination of standard ScenarioMIP
experiments (out to 2100) augmented with the first 25 years of the extensions (out to 2125). Several sectors have clear interest
in longer-term scenarios out to 2500 (e.g., for modeling inert systems such as forest ecosystems within the terrestrial
biodiversity sector and sectors at risk of impacts from cryospheric changes and sea level rise).
The growing scientific capabilities and public interest in detection and attribution put a spotlight on I&A outputs from
the Detection and Attribution MIP (DAMIP) given direct implications for risk management, which also indicates applications
within the Pre-industrial (Pi)Control experiment. Finer resolution outputs are also a shared interest of the I&A Theme,
HighResMIP (Roberts et al., 2025) and the Decadal Climate Prediction Project (DCPP) (Boer et al., 2016). MIP experiments
with relevance to a particular sector or dynamical process may be of particular interest (e.g., the Land Use Model
Intercomparison Project (LUMIP) (Lawrence et al., 2016) for some agriculture and ecosystem variable groups) but are not as
broadly used for application. There is also general interest in determining the sensitivity of climate impacts to uncertain ESM
components, but this is at the discretion of the MIP leaders and we would recommend one or a very small number of MIP
experiments producing extensive I&A variable groups beyond the historical and ScenarioMIP experiments (as resources
allow).

**5.1.4 Time subsets**
ESM groups with resource constraints may elect to produce more extensive I&A variable groups for a limited time period, or
time subset(s), rather than the entire historical and 21st century periods. This saves resources and provides I&A outputs that
would not be possible if there was a requirement to produce all variables for all years, but begs the question as to which time
subsets would be most desirable for VIACS communities. Engagement by the I&A Data Request Theme found greatest
interests in understanding the climate of the most recent past, with 10 years likely the minimal number of years required to
calculate important climate statistics even as 30 years matches WMO guidance for climatological analysis (WMO, 2017),
More rare events typically benefit from longer climatological normals, although this is not always practical in a non-stationary
climate. IPCC AR6 used 20-year time subsets to represent distinct global warming levels, with that time span also being more
relevant within a non-stationary climate given that climate shifts are occurring on similar time horizons (IPCC, 2021). Time
subsets for the recent past also benefit from alignment with modern-era satellite observations and retrospective analyses
(Gelaro et al., 2017), for example the 1979-2025 period. Projections for the next 10 years are likely to only have a statistically-
weak difference from the previous decades, so we recommend using future time subsets centered first on the 2050s (when
many climate signals emerge) followed by a late century time subset centered on the 2080s (when scenarios are more distinct)
(Ranasinghe et al., 2021). These time subsets also align with decision processes supported by the VIACS communities (Stuart
et al., 2024). As in the historical period, future time subsets of 20, 30, or more years are helpful. The team also considered
time subsets centered around targeted global warming levels (GWLs); however, this is less practical given that it would require
an iterative process to run initial ESM simulations that determine the period corresponding to a given GWL, then re-running
the ESM with more extensive I&A outputs in that time subset. Additionally, not all variables and climatic impact-drivers
cleanly track GWLs (Tebaldi et al., 2024).
As an illustration of one sector's use of different time subsets, energy planning decisions are usually based on
prospective studies targeting the next 5-10 years, although planning efforts are increasingly reaching 20-25 years. Those
prospective studies generally rely on recent climate conditions, using 30-year time series of the required variables coming from
station observations or reanalyses, representative of the current climatology (1991-2020 currently for example). Continuing
climate change means that the recent past is no longer sufficient to anticipate even near future climate evolution, especially for
heat wave frequency. Climate projections can provide a large scope of possible combinations of climate response to the human
induced radiative forcing and decadal variability; however, decadal predictions may help reduce/specify this ensemble of
possibilities for the next 5-10 years.

**5.1.5 Ensemble members and gap filling**
Robustness of ESM output for impacts and adaptation applications may also be improved by driving multiple initial condition
ensemble members for a given experiment. This is particularly intriguing for VIACS applications given that initial condition
ensembles provide more simulated years in a given climatological period or GWL within a non-stationary climate system that
features a relatively larger range of internal variability than the climate change signal itself (Chan et al., 2022). VIACS
applications manage uncertainty across ESM ensemble members, ESM groups, and model versions and physics packages
within a given ESM. For the latter category, perturbed physics ensembles can provide important information about parameter
uncertainty and model stability; however, these are of less interest for extensive I&A outputs given the likely high data volume.
Large ensemble member experiments are particularly useful for infrequent extreme events that are at the forefront of impacts
and adaptation concerns. When resources are constrained, artificial intelligence and other statistical approaches may help us
fill in gaps when the full set of years, variables, time resolutions, ensemble members and experiments are not practical. It may
therefore be helpful to simulate (at least) a subset of model ensemble members in a targeted time subset with more
comprehensive variable outputs and sub-daily temporal resolution. These could then serve as a training dataset to emulate
other ensemble members, models, time subsets, variables and time resolutions building off a less resource-intensive set of
background simulations (e.g., monthly outputs of core variables over more years, models and ensemble members).

**5.2 Gaps and opportunities across impacts and adaptation applications**
The I&A Data Request process led to the identification of several gaps in earth system processes that support climate
applications. While this effort has focused on the ESM component of the applications chain outlined in Fig. 1, in some cases
the bottleneck for applications comes further down. The provision of impacts and adaptation ESM variables opens the door to
many opportunities for research and improvement all along the chain. Downscaling is strongly influenced by the selection of
RCMs, as well as resources required to operate high-resolution nested models spanning the world. Bias adjustment is built on
a foundation of high quality and long period Earth observations that are not available for many impacts-relevant variables.
Improved ESM resolution and process understanding will reduce the need for downscaling and bias-adjustment to reach current
application scales, but will enable reach to even finer decision-relevant scales.  Impacts models have their own biases and
uncertainties and most can only represent responses to a subset of the climatic impact-drivers (Wang et al., 2024). Improved
impacts models will therefore better translate climatic impact-drivers into projected impacts for natural and human systems,
as well as allow evaluation of adaptation interventions.
Additional requested variables proved too challenging for the first version of the I&A Data Request given required
shifts in model infrastructure and requirements around constructed indices, but these remain of great interest in the community
and motivate future ESM capability improvements. To reduce the burden of high-resolution outputs, there remains a potential
to save less frequent outputs that capture high-frequency statistics and characteristics.  As noted also in the previous CMIP
VIACS request, the impacts and adaptation community would benefit from ESM groups saving monthly outputs of event count
or statistical distribution parameters related to daily or sub-daily events (Ruane et al., 2016), for example monthly counts of
the number of rainy days, hourly precipitation rates exceeding a threshold, or the number of frost events. For CMIP7 there was
an additional request for data that would provide information about the daily timing of extreme events, for example noting the
time of day when maximum temperature was simulated. Additional variables like heat indices requested by the Health Impacts
opportunity could be assembled by the applications expert if all ingredients to the index's numerical recipe are provided, but
these indices were requested to reduce the burden on output users and also to minimize the likelihood that errors or slight
methodological differences could affect applications.  Calculating these variables within the ESMs also allows more efficient
outputs, for example the maximum daily wet bulb globe temperature (WBGT) is a single key impacts relevant variable that
spares modeling groups from having to provide its many individual variable components at hourly resolution.  The I&A Team
thus encourages models to develop new code and structures to make these outputs feasible, and encourages practitioners and
data providers to share best practices, tutorials and guidelines as part of broader capacity building for CMIP7 applications. For
example, we encourage continuing collaborations around CMIP7 data provision, access, further processing and applications
between CMIP and the World Climate Research Programme's Regional Information for Society (RIfS) Joint Task Team on
Responsible Data Use, as well as with the World Adaptation Science Program (WASP).
I&A requests for ESM outputs also set the stage for the rapidly evolving community of climate services. Recent
efforts have increased the utility of climate information for many applications, including efforts to work directly with
professional societies such as the International Actuarial Association working to manage changing insurance baselines in the
face of rising seas and increased risk of coastal floods, fire weather, flooding, drought and heat waves (Connors et al., 2022).

## 5.3 Key reflections from data request process

The Impacts and Adaptation data request is unique within CMIP7 in having a mandate to reach far beyond the ESM groups
that are at the center of the CMIP community. The web of applications stemming from CMIP work extends to a huge number
of experts, policymakers, stakeholders, and practitioners across all aspects of society.  This is truly a testament to the impact
and ongoing legacy of the CMIP endeavor, and it is likely under-appreciated even by many who are directly involved. The
VIACS communities are distributed and have unique networks and collaboration structures, and they are eager to be engaged.
CMIP7 has demonstrated a growing recognition of the importance of the VIACS communities; not just as model
users, but as participants with helpful input into all phases of planning, conduct, sharing, and interpretation of results.
Continuity of the VIACS Advisory Board between CMIP6 and CMIP7, as well as VIACS participation in CMIP panel
meetings and workshops, helped ensure a continuation of institutional experience and networks that allowed the I&A Data
Request Team to hit the ground running in recruitment, goal-setting, outreach, and translation of inputs into specific and
practical data requests.  The I&A Team also benefited from an open nomination and selection process that emphasized broad
perspectives and strong networks to application communities, drawing from the VIACS Advisory Board but extending far
beyond.
CMIP7 enables a process for community input and ESM group uptake that would not be feasible through independent
outreach. I&A Team and CMIP7 IPO efforts to make the input solicitation process clear, accessible and easy through
established and informal network outreach and the Mural Board proved invaluable given that CMIP was requesting voluntary
contributions from busy community experts. The I&A data request author team, participating in a voluntary basis, also
appreciated efforts by the CMIP7 Data Request Task Team participants to track details and translate technical information
between subsequent versions of the AirTable.
CMIP7 also supports climate services and decision making beyond the information generation. Outreach must be
cognizant of stakeholder fatigue, as each new round of projections and the subsequent updates to scientific literature do not
necessarily match the investment or policy development cycles in private and public decision making. The Rapid Evaluation
Framework and Diagnostic, Evaluation, and Characterization of Klima (DECK) Experiment Group analyses are critical to
helping stakeholders build familiarity with the new sets of outputs, helping to motivate capacity investments and new analyses
required to switch from planning based around CMIP5 or CMIP6 to planning based upon the state-of-the-science CMIP7
(Dunne et al., 2025). Improvements in accuracy and changes to the baseline context (e.g., land use changes and emissions of
greenhouse gases and aerosols) would be particularly compelling in justifying stakeholders' shift to planning with the latest
CMIP7 outputs and scenarios. CMIP7 and climate service organizations can increase uptake by making outputs more
accessible. Examples may include the provision of outputs in decision-ready format that is directly ingestible by catastrophe
models and optimization analysis, or pre-computed ensemble statistics that would allow users, such as practitioners in the
financial sector, to consistently apply climate related impacts in a transparent and reproducible manner. CMIP7 efforts to
standardize data taxonomy, spatial and temporal resolution, and metadata are therefore critical to early adoption.

## 6 Conclusions

The I&A Data request process aimed to establish consensus on output needs and priorities across a broad spectrum of ESM
output users in the VIACS communities and the ESM groups, resulting in 13 I&A data request opportunities, 60 variable
groups (Section 4) and 539 unique variables (Appendix B).  The various opportunities and variable groups discussed here can
inform decisions impacting systems across society and the natural world even as these systems are increasingly intertwined.
This process required collaboration between climate scientists and impacts sector practitioners, with experts who can converse
in both spheres providing especially helpful insights.
Throughout this data request the I&A Thematic team has endeavored to emphasize how different levels of resource
commitments can maximize the impact of CMIP7 for some of society's most pressing challenges.  This includes minimal or
core variable groups that are likely to enable broad use (Section 4), the identification of smaller subsets of variables around
targeted applications (Section 4), and time subset prioritization for focused analysis (Section 5). The I&A data request also
invites ESM groups with more resources to drive larger scientific advances by meeting the rising demand for more variables,
new variables, and high-resolution variables that can allow applications to match the complexity of the systems they aim to
protect and help thrive. The overall process is enlightening for CMIP leadership, modeling groups producing ESM outputs,
and ESM development priorities that will elevate the application and utility of ESM outputs. Even those groups that cannot
meet this full CMIP7 request may consider ways to be better positioned to meet I&A requests for CMIP8.

Pressing deadlines from policymakers and an increasing appetite for systems-level knowledge emphasize the need

for clear organization and communication of CMIP7 governance, data processing and dissemination of ESM outputs. This is
especially true for impacts and adaptation applications that require additional information processing in the form of
downscaling, bias-adjustment, impacts modeling, analyses, and decision making around implementation. Waiting to get perfect
ESM outputs may leave these later phases of application without enough time to properly inform policymakers and
stakeholders through initiatives such as the IPCC. The Rapid Evaluation Framework may help facilitate this process. It would
be beneficial for CMIP7 modeling groups to provide clear communication on variable groups they intend to produce, allowing
important preparation to set the stage for CMIP7 across the impact & adaptation sectors. Engagement up and down that pipeline
is important to indicate priorities, communicate uncertainties, track error propagation, and give due credit to scientific
contributors who may not be at the stakeholder interface.

The I&A Team also received feedback that clear documentation of outputs is needed to facilitate broad use in

applications. Lack of clarity or barriers to use would create the need for more intermediary organizations (e.g., consulting
groups) that can navigate this space, but that would increase the distance between those who produce the data and those who
use it, with the potential to lead to information degradation or misuse (particularly in the Global South). Public-private-
academic collaborations hold great potential to provide open climate services for research that would enable a large number
of applications and support pressing decisions around the world. The I&A Team thus encourages ESM teams to follow through
and stretch to meet the needs of this important community, which carries key climate messages to the world with an emphasis
on the specific points of contact whereby climate change affects the things that society cares about.
**Appendix A. Data Request Opportunity Evaluation and Processing**

**Table A.1:** Actions taken in the processing of proposed opportunities for the I&A Data Request Theme. The processing of
opportunities proposed in the open call of August 2024, including proposals from both within the author team and more widely,
was carried out by revising the evaluation made within each thematic author team in the framework of a cross-thematic meeting
in mid-September 2024. The meeting participants selected certain opportunities, rejected some, and merged others with shared
scientific objectives and domain. In a subsequent step, an interactive discussion was held between members of our author team,

 opportunity proposal leaders, and the relevant domain communities. The goal was to harmonize the initially proposed
opportunities and improve their description and data requirements. The following table summarises the key processing actions
and decisions with specific reference to a working copy Airtable database available at the following link https://bit.ly/CMIP-
DR-Opportunities.

| ID | Meeting decision made | Notes from consultation | Notes from Author team |
|---|---|---|---|
| **Accepted** | | | |
| ID 5: Understanding the role of atmospheric composition for air quality and climate change | Author team meeting 2024-11-11 | Checked with the Impacts and Adaptation theme if health impacts should be included here, or in a dedicated Opportunity. Decided to proceed with a dedicated health impacts Opportunity (ID 37, see Ruane et al., 2025). Discussion needed to ensure consistent pressure levels across Data Request. | Concern that all variable groups are listed as 'High priority'. In response, proposers highlighted that priorities were determined by the AerChemMIP community. Title was also shortened following review. |
| ID 11: Terrestrial Biodiversity | Author team meetings 23-09-2024 and 24-09-2024 | Agreed to merge ID 2 (Advanced process-explicit forest models), ID 3 (Advanced process-explicit terrestrial biodiversity models), and ID 4 (Advanced high-resolution biophysical models for terrestrial biodiversity) into ID 11(BES-SIM standard scenarios for terrestrial biodiversity and ecosystem services) and rename. | Combination of several opportunities covering terrestrial biodiversity and ecosystems |
| ID 16: Climate impacts on marine biodiversity and ecosystems | Cross thematic review, October 2024 | 23/24-09-24: merge ID 21, 33, and possibly 44 and 16. Phase 1 consultation: merge ID 21, 33, 32 and 16. Cross thematic review (October 24): merge ID 21, 33, 32 into 44. | Combination of several opportunities covering marine ecosystems and fisheries. In the end, IDs 21, 32 and 33 were merged with ID44 rather than with ID 16. |
| ID 19: Agriculture and Food System Impacts | Author team meetings 23-09-2024 and 24-09-2024 | Agreed to merge ID 7 (Agricultural water requirements), and ID 30 (Enhanced agricultural impacts) into ID 19 (Core agricultural modeling) and rename. ID 6 (Agricultural carbon modeling) was also merged into the opportunity. | Includes opportunities that were merged into variable groups for agricultural modeling, water resources, air pollution, and carbon in agricultural systems. |
| ID 20: Core climate services | | | High resolution (sub-daily) outputs for use in multiple climate services applications across sectors. |
| ID 22: Energy system impacts | Author team meetings 23-09-2024 and 24-09-2024 | Agreed to merge ID 25 (Damaging weather for energy systems), ID 45 (Modeling energy demand in built environment), ID 57(Risk assessment for off-shore wind farm installation), and ID 58 (River temperature modelling for thermal power plants) into ID 22(Core variables for energy system) and rename. | Combination of different opportunities devoted to energy system planning and adaptation in balancing generation and demand and ensuring resilience to extreme weather conditions |
| ID 24: Advancing Wind Wave Climate Modelling for Coastal Zone | Author team meeting 19-09-2024 | Noted need for high frequency surface conditions. Suggestion to merge ID 56 (Risk Assessment for offshore wind farm | ID 56 merged into this opportunity. ID 68 to remain separate (see below). |

| | | installation and ID 68 (Wind driven Ocean Surface Waves) - see below. | |
|---|---|---|---|
| Dynamics, Impacts, and Risk Assessment | | | |
| ID 26: Detection and Attribution | Author team meeting 2024-11-11 | Opportunity name revised to be more specific. Suggestion to add more biogeochemical variables and ocean grid variables. | Discussion needed to ensure consistent pressure levels across Data Request. |
| ID 31: Role of fire in the Earth system | Author team meeting 24-11-2024 | A key Earth System Process and emerging area of science. Need to add more atmospheric variables. | A more descriptive title for this opportunity should be provided. Variable list may be reduced to focus more on fire related parameters. |
| ID 35: Glacier changes, drivers, and impacts | Author team meeting 25/10/2024 | The cross-thematic review suggested investigating the need for a more specific variable group. It was also suggested to include mountain glaciers in the title and the possibility of merging with Hydrology and Fresh Water Systems (ID 38), Water Resources (ID 67) and Water Cycle (ID 66), which could be consolidated into two Opportunities. | Discussion of potential merger and decision to retain a distinct glacier Opportunity and to re-name (from original Glacier geometry changes and glacio-hydrology). |
| ID 37: Health impacts | Author team meetings 23-09-2024 and 24-09-2024 | Merge ID 8 (Air pollution health impacts), ID 34 (Flooding health impacts) into ID 37 (Health impacts 2) and rename. | Combines multiple opportunities as variable groups related to heat and flooding, including daily and sub-daily outputs as well as heat indices. |
| ID 38: Assessments for Hydrological Processes, Water Resources, and Freshwater Systems | Author team meeting 25/10/2024 | The cross-thematic review suggested consolidating this Opportunity (previously Hydrology and Freshwater Systems), the Water Resources (ID 67) and Water Cycle (ID 66) into two Opportunities, where one is focused on the water cycle and the other on land/water resources. The proposers of all three Opportunities were requested to make the necessary edits and a merge with ID 67 and Climate impact assessments on freshwater ecosystems (ID 15) was requested. | After consideration of potential for proposed merges, author team decided, together with input from Impacts & Adaptation team that ID 67 (see below) be merged while ID 66, in discussion with and led by the Earth System team, should remain standalone as it is a fundamental component together with energy and carbon cycles. A soil temperature profile was also identified to be included for permafrost. |
| ID 40: Vulnerability of urban systems, infrastructure and population | Merge recorded in the I&A "Opportunities Being Merged" document, issued 19-09-2024 | Merge ID 14 (Cities) into ID 40 (Impact of climate variability and change on urban population and infrastructure). | Combines several cities and infrastructure opportunities. |
| ID 41: Impacts of climate change on aviation | Discussed in author team meeting 19-09-2024 | Identified as a high-volume data request due to misinterpretation of requested levels. | A specific set of levels (UTLS: upper troposphere/lower stratosphere) was created for the corresponding variable groups in March 2025. |
| ID 42: Impacts of climate change on transport infrastructure | Discussed in author team meeting 19-09-2024 | | Combines data requests from users focused on impacts on road and rail infrastructure. |
| ID 44: Changes in marine biogeochemical cycles and ecosystem processes | Author team meeting 02-10-2024 https://github.com/CMIP-Data-Request/Harmonised-Public-Consultation/issues/32 | Potential to merge with other BGC opportunities - author team following up with proposers. | Revise original name, include additional marine variable groups and merge in other opportunities. |

| | | | |
|---|---|---|---|
| ID 47: Ocean Changes, Drivers and Impacts | Author sub-group meeting 03-02-2025 | Deferred to thematic team pending further variables inclusion. Suggestion to merge ID 56 (see below). ID56 was merged with ID24. | Variable groups confirmed, refined and new variables added. |
| ID 49: Ocean extremes | Author team meeting 06-11-2024 | Deferred to thematic team for further variable development and inclusion and merge with ID 62 (SurgeMIP storm surge intercomparison - see below) | Merge with ID62 completed and new physical parameters (and associated variables) added. |
| ID 54: Multi-annual-to-decadal predictability of the Earth System and risk assessment of climate extremes | Author team meeting 24-11-2024 | This request has selected the entire fast track. If the data volume is a problem, I expect this can probably be narrowed down to a more refined set of experiments that are of most relevance. | This opportunity involves a broad list of variables, which may be split into separate and smaller groups by redesigning in favour of a subset of more specific opportunities. |
| ID 55: Rapid Evaluation Framework | Author team meeting 2024-11-11 | Required variables confirmed across all themes following discussion. Opportunity ID 23 merged into this. | Added DECK experiment group and coordinated with Model Benchmarking TT on confirmed diagnostics for inclusion. |
| ID 63: Synoptic systems | Author team meeting 2024-11-11 | Title changed to remove 'impacts' to avoid confusion with I&A theme work. | Variables gathered through community consultation. |
| ID 64: Diagnosing temperature variability and extremes | Author team meeting 2024-11-11 | Suggestions to merge with other 'extremes' relevant Opportunities. | Decided not to merge, but title and description updated to make distinction clear. |
| ID 65: Uncertainty in changing net primary production | Author team meeting 27-09-2024 Merge in ID 44 | Potential to merge with other BGC opportunities (changing net primary production, biological carbon sink in ocean and Marine BGC) | This could be coordinated/merged with the others marine biogeochemical opportunities. |
| ID 67: Water security and Freshwater Ecosystem Services | Author team meeting 21-11-2024 | Merge ID 15 (Climate impact assessments on freshwater ecosystems) into ID 67 (Water Resources (Hydroclimatic hazards at regional scale for adaptation actions and resilience building.)) and rename. | Combines two opportunities with overlapping objectives involving freshwater resources |
| ID 68: Effects and Feedbacks of Wind-Driven Ocean Surface Waves Coupled Within Earth System Models | Author sub-group meeting 03-02-2025 | Deferred to thematic team with suggestion to merge with ID 24 | ID 24 concerns offline wave models, whereas the focus here is on coupled ESM wave components. Opportunity title updated from original Wind driven Ocean Surface Waves to reflect. |
| ID 69: Baseline Climate Variables for Earth System Modelling | | | Created for all data request with cross-cutting relevance to all themes. |
| ID 77: Bias-adjustment for impacts modelling and analysis | Author team meetings 23-09-2024 and 24-09-2024 | Merge ID 17 (Climatologies at high resolution for the earth's land surface areas - CHELSA) into ID 77 (ISIMIP Inter-Sectoral) and rename. | Includes opportunities originating from CHELSA and ISIMIP. Daily deposition variables. |
| ID 80: Empirical Statistical Downscaling and Emulators | | | Key variables needed for both traditional ESD methods and newer hybrid emulators. With ID82, responds to increasing user demand for high-frequency climate information. |
| ID 81: Dynamical Downscaling | Recorded in update to the I&A Opportunities Being Merged document 09-10-2024 | Merge ID 27 (Data for dynamical downscaling) and ID 28 (Atmospheric and Oceanic | Combines several opportunities independently requesting variable groups associated with higher |

| | | Dynamical downscaling) into ID 81 (Dynamical Downscaling). | resolution modeling of the atmosphere and ocean. |
|---|---|---|---|
| ID 82: Robust Risk Assessment of Tipping Points | Author team meeting 24-11-2024 | TipMIP set of experiments will be added once available on the controlled vocabularies system. | The objectives are clear and the proposed set of variables is consistent. |
| **Merged** | | | |
| ID 2: Advanced process-explicit forest models | Author team meetings 23-09-2024 and 24-09-2024 | | Merged into the opportunity Terrestrial Biodiversity (ID 11) to group this impact sector |
| ID 3: Advanced process-explicit terrestrial biodiversity models | Author team meetings 23-09-2024 and 24-09-2024 | | Merged into the opportunity Terrestrial Biodiversity (ID 11) to group this impact sector. |
| ID 4: Advanced, high-resolution biophysical models for terrestrial biodiversity | Author team meetings 23-09-2024 and 24-09-2024 | | Merged into the opportunity Terrestrial Biodiversity (ID 11) to group this impact sector. |
| ID 6: Agricultural carbon monitoring | Author team meetings 23-09-2024 and 24-09-2024 | | Merged into Agriculture and Food System Impacts Opportunity (ID 19) to group alike requests. |
| ID 7: Agricultural water requirements | Author team meetings 23-09-2024 and 24-09-2024 | | Merged into Agriculture and Food System Impacts Opportunity (ID 19) to group alike requests. |
| ID 8: Air pollution health impacts | Author team meetings 23-09-2024 and 24-09-2024 | | Merged into Health Impacts Opportunity (ID 37) to group alike requests. |
| ID 14: Cities | Merge recorded in the I&A "Opportunities Being Merged" document, issued 19-09-2024 | | Merged into Vulnerability of urban systems, infrastructure and population (ID 40) to group alike requests. |
| ID 15: Climate impact assessments on freshwater ecosystems | Author team meeting 21-11-2024 | | Merged into Water security and Freshwater Ecosystem Services Opportunity (ID 67) to group alike requests. |
| ID 17: Climatologies at high resolution for the earth's land surface areas - CHELSA | Author team meetings 23-09-2024 and 24-09-2024 | | Merged into Bias-adjustment for impacts modeling and analysis Opportunity (ID 77) to group alike requests. |
| ID 21: Core fisheries modeling output | Author team meeting 27-09-2024 Merge in ID 44 | Request proposers to merge with Fisheries board on additional fisheries modeling and impacts (ID32) and Fisheries board on advanced mariculture | Integrate with companion opportunities into joined Changes in marine biogeochemical cycles and ecosystem processes opportunity (ID 44) |
| ID 25: Damaging weather for energy systems | Author team meetings 23-09-2024 and 24-09-2024 | | Merged with other opportunities into the energy system impacts opportunity (ID22) |
| ID 27: Data for dynamical downscaling | Recorded in update to the I&A Opportunities Being Merged document 09-10-2024 | | Merged into the opportunity Dynamical Downscaling (ID 81) to group alike requests. |
| ID 28: Atmospheric and Oceanic Dynamical downscaling | Recorded in update to the I&A Opportunities Being Merged document 09-10-2024 | | Merged into the opportunity Dynamical Downscaling (ID 81) to group alike requests. |
| ID 30: Enhanced agricultural impacts | Author team meetings 23-09-2024 and 24-09-2024 | | Merged into Agriculture and Food System Impacts Opportunity (ID 19) to group alike requests. |
| ID 32: Fisheries board on additional fisheries modeling and impacts | Author team meeting 27-09-2024 Merge in ID 44 | Request proposers to merge with Core fisheries modeling output (ID21), and Fisheries board on | Integrated with companion opportunities (ID21 and ID 33) into ID 44. |

| | | | |
|---|---|---|---|
| | | advanced mariculture and species model (ID33) into one Opportunity. | |
| ID 33: Fisheries board on advanced mariculture and species model | Author team meeting 27-09-2024 Merge in ID 44 | Request proposers to merge with Core fisheries modeling output (ID21) and Fisheries board on additional fisheries modeling and impacts (ID32) into one Opportunity. | Integrated with companion opportunities (ID21 and ID32) into ID 44. |
| ID 34: Flooding health impacts | Author team meetings 23-09-2024 and 24-09-2024 | | Merged into Health Impacts Opportunity (ID 37) to group alike requests. |
| ID 45: Modelling energy demand in built environment | Author team meetings 23-09-2024 and 24-09-2024 | | Merged with other opportunities into the energy system impacts opportunity (ID22) |
| ID 56: Risk assessment for offshore wind farm installation | Author team meeting 27-09-2024 Merge in ID24 | Deferred to thematic team with suggestion to merge with ID 24 | Merged into Merged into Ocean and Sea Ice Theme, Advancing Wind Wave Climate Modelling for Coastal Zone Dynamics, Impacts, and Risk Assessment opportunity (ID 24) to group alike requests. |
| ID 58: River temperature modelling for thermal power plants | Author team meetings 23-09-2024 and 24-09-2024 | | Merged with other opportunities into the energy system impacts opportunity (ID22) |
| ID 59: Robust risk assessments of extreme climate | Cross thematic review, October 2024 | Merge with ID 54 (Multi-annual-to-decadal predictability of the Earth System and risk assessment of climate extremes). | Merged with ID 54 |
| ID 62: SurgeMIP (storm surge intercomparison) | Cross thematic review, October 2024 | Merge with ID 49 (Ocean Extremes) | Merged with Ocean Extremes (ID 49) |
| **Rejected** | | | |
| ID 36: Heat Health Impacts | | | Initial submission contained an error which led to an improved submission that is now included in Heat Impacts Opportunity (ID 37) |



**Appendix B. Description of new variables proposed by Impacts and Adaptation Data Request Theme**

**Table B.1**: New variables proposed and variable formula updates by the Impacts and Adaptation Data Request Theme. The variables that are newly introduced in CMIP7 are tabulated below. The Coordinate Specifications column is lists special aspects of the temporal and spatial requirements for each variable. The full grid specifications can be found in v1.2 of the CMIP7 Data Request (Data Request Task Team, 2025a).

| Variable CMOR name | CF standard name | Description | Further detail to aid compute** | Coordinate specifications |
|---|---|---|---|---|
| bldep | atmosphere_boundary_layer_thickness | Boundary Layer Depth | existing physical parameter | 3hrPt (3-hourly), E1hr (hourly) |
| cfc114 | mole_fraction_of_cfc114_in_air | mole fraction of cfc114 | | AERmon (monthy) |
| clt | cloud_area_fraction | Total Cloud Cover Percentage | existing physical parameter | E1hr (hourly) |
| drynh3 | minus_tendency_of_atmosphere_mass_content_of_ammonia_due_to_dry_deposition | Dry Deposition Rate of NH3 | existing physical parameter | AERday (daily) |
| drynh4 | minus_tendency_of_atmosphere_mass_content_of_ammonium_dry_aerosol_particles_due_to_dry_deposition | Dry Deposition Rate of NH4 | existing physical parameter | AERday (daily) |
| drynoy | minus_tendency_of_atmosphere_mass_content_of_noy_expressed_as_nitrogen_due_to_dry_deposition | Daily Dry Deposition Rate of NOy | existing physical parameter | AERday (daily) |
| emich4 | tendency_of_atmosphere_mass_content_of_methane_due_to_emission | total emission rate of CH4 | | AERmon (monthly) |
| epc1000 | sinking_mole_flux_of_particulate_organic_matter_expressed_as_carbon_in_sea_water | Downward Flux of Particulate Organic Carbon at 1000m | | Omon (monthly) |
| epcalc1000 | sinking_mole_flux_of_calcite_expressed_as_carbon_in_sea_water | Downward Flux of Calcite at 1000m | | Omon (monthly) |
| expcob | sinking_mole_flux_of_particulate_organic_matter_expressed_as_carbon_in_sea_water | Sinking Flux of Particulate Organic Carbon Reaching the Ocean Bottom | | Oday (daily) |
| ficeberg | water_flux_into_sea_water_from_icebergs | Water Flux into Sea Water from Icebergs | existing physical parameter | 3hr (3-hourly) |
| flandice | water_flux_into_sea_water_from_land_ice | Water Flux into Sea Water from Land Ice | existing physical parameter | 3hr (3-hourly) |
| flashrate | frequency_of_lightning_flashes_per_unit_area | Lightning Flash Rate | existing physical parameter | Eday (daily) |
| friver | water_flux_into_sea_water_from_rivers | Water Flux into Sea Water from Rivers | existing physical parameter | Oday (daily), 3hr (3-hourly) |
| hcfc22 | mole_fraction_of_hcfc22_in_air | Mole Fraction of HCFC22 | | AERmon (monthly) |
| hfc125 | mole_fraction_of_hfc125_in_air | Mole Fraction of HFC125 | | AERmon (monthly) |
| hfc134a | mole_fraction_of_hfc134a_in_air | Mole Fraction of HFC134a | | AERmon (monthly) |
| hfds | surface_downward_heat_flux_in_sea_water | Downward Heat Flux at Sea Water Surface | existing physical parameter | 3hr (3-hourly) |
| hfls | surface_upward_latent_heat_flux | Surface upward latent heat flux | existing physical parameter | E1hr (hourly) |
| hfrunoffds | temperature_flux_due_to_runoff_expressed_as_heat_flux_into_sea_water | Temperature Flux Due to Runoff Expressed as Heat Flux into Sea Water | existing physical parameter | 3hr (3-hourly) |

| hfss | surface_upward_sensible_heat_flux | Surface upward sensible heat flux | existing physical parameter | E1hr (hourly) |
|------|------|------|------|------|
| hurs | relative_humidity | Relative humidity at 2m above the surface | existing physical parameter | E1hr (hourly) |
| huss | specific_humidity | Specific humidity at 2m. | existing physical parameter | E1hr (hourly) |
| intpoc | ocean_mass_content_of_particulate_organic_matter_expressed_as_carbon | Particulate Organic Carbon Content | existing physical parameter | Oday (daily) |
| intpp | net_primary_mole_productivity_of_biomass_expressed_as_carbon_by_phytoplankton | Primary Organic Carbon Production by All Types of Phytoplankton | existing physical parameter | Oday (daily) |
| intppnano | net_primary_mole_productivity_of_biomass_expressed_as_carbon_by_nanophytoplankton | Vertically integrated total primary (organic carbon) production by nanophytoplankton. | | Omon (monthly) |
| irrDem | surface_downward_mass_flux_of_water_due_to_irrigation | irrigation water demand | | day (daily) |
| irrGw | surface_downward_mass_flux_of_water_due_to_irrigation | irrigation water withdrawal from groundwater | | day (daily) |
| irrLut | surface_downward_mass_flux_of_water_due_to_irrigation | irrigation water withdrawal | existing physical parameter | day (daily) |
| irrSurf | surface_downward_mass_flux_of_water_due_to_irrigation | irrigation water withdrawal from surface water | | day (daily) |
| mlotst | ocean_mixed_layer_thickness_defined_by_sigma_t | Ocean Mixed Layer thickness Defined by Sigma T | existing physical parameter | Oday (daily) |
| mrro | runoff_flux | Total Runoff | existing physical parameter | Eday (daily) |
| mrso10 | mass_content_of_water_in_soil_layer | Soil moisture in the top 10 cm of the soil column | | Eday (daily) |
| noaahi2m | heat_index_of_air_temperature | mean 2m daily NOAA heat index | | day (daily) |
| noaahi2mmax | heat_index_of_air_temperature | max 2m daily NOAA heat index | | day (daily) |
| o2200 | mole_concentration_of_dissolved_molecular_oxygen_in_sea_water | Dissolved Oxygen Concentration at 200 meters | existing physical parameter | Oday (daily) |
| o2min | mole_concentration_of_dissolved_molecular_oxygen_in_sea_water_at_shallowest_local_minimum_in_vertical_profile | Oxygen Minimum Concentration | existing physical parameter | Oday (daily) |
| o2os | mole_concentration_of_dissolved_molecular_oxygen_in_sea_water | Surface Dissolved Oxygen Concentration | existing physical parameter | Oday (daily) |
| ph200 | sea_water_ph_reported_on_total_scale | pH, negative log10 of hydrogen ion concentration with the concentration expressed as mol H kg-1. | existing physical parameter | Oday (daily) |
| phyc200 | mole_concentration_of_phytoplankton_expressed_as_carbon_in_sea_water | Phytoplankton Carbon Concentration | existing physical parameter | Oday (daily) |
| phycalc | mole_concentration_of_calcareous_phytoplankton_expressed_as_carbon_in_sea_water | Mole concentration of calcareous phytoplankton expressed as carbon in sea water | existing physical parameter | Oday (daily) |
| Phydiat | mole_concentration_of_diatoms_expressed_as_carbon_in_sea_water | Carbon concentration of diatoms | Mole Concentration of Diatoms Expressed as Carbon in Sea Water | Oday (daily) |
| phydiaz | mole_concentration_of_diazotrophic_phytoplankton_expressed_as_carbon_in_sea_water | Carbon concentration of diazotrophs | Mole Concentration of Diazotrophs Expressed as Carbon in Sea Water | Oday (daily) |
| phymisc | mole_concentration_of_miscellaneous_phytoplankton_expressed_as_carbon_in_sea_water | Carbon concentration of miscellaneous phytoplankton | Mole Concentration of Miscellaneous Phytoplankton | Oday (daily) |

| | | | Expressed as Carbon in Sea Water | |
|---|---|---|---|---|
| phynano | mole_concentration_of_nanophytoplankton_expressed_as_carbon_in_sea_water | Carbon concentration of nanophytoplankton | Mole Concentration of Nanophytoplankton Expressed as Carbon in Sea Water | Oday (daily) |
| phypico | mole_concentration_of_picophytoplankton_expressed_as_carbon_in_sea_water | Carbon concentration of picophytoplankton | Mole Concentration of Picophytoplankton Expressed as Carbon in Sea Water | ODay (daily) |
| Ps | surface_air_pressure | Surface pressure | existing physical parameter | E1hr (hourly) |
| Rlds | surface_downwelling_longwave_flux_in_air | surface_downwelling_longwave_flux_in_air | existing physical parameter | E1hr (hourly) |
| rlus | surface_upwelling_longwave_flux_in_air | surface_upwelling_longwave_flux_in_air | existing physical parameter | E1hr (hourly) |
| rsdo | downwelling_shortwave_flux_in_sea_water | Downwelling Shortwave Radiation in Sea Water | existing physical parameter | Oday (daily) |
| rsds | surface_downwelling_shortwave_flux_in_air | downward solar radiation flux at the surface | existing physical parameter | E1hr (hourly) |
| rsdsdiff | surface_diffuse_downwelling_shortwave_flux_in_air | Surface Diffuse Downwelling Shortwave Radiation | existing physical parameter | E1hr (hourly) |
| rsntds | net_downward_shortwave_flux_at_sea_water_surface | Net Downward Shortwave Radiation at Sea Water Surface | existing physical parameter | 3hr (3-hourly) |
| rsus | surface_upwelling_shortwave_flux_in_air | Surface upwelling shortwave radiation | existing physical parameter | E1hr (hourly) |
| sfcWind | wind_speed | surface wind speed | existing physical parameter | E1hr (hourly) |
| sfdsi | downward_sea_ice_basal_salt_flux | Downward Sea Ice Basal Salt Flux | existing physical parameter | 3hr (3-hourly) |
| sfpm1 | mass_fraction_of_pm1_ambient_aerosol_particles_in_air | PM1.0 mass mixing ratio in lowest model layer | | AERday (daily), AERhr (hourly) |
| sfpm10 | mass_fraction_of_pm10_ambient_aerosol_particles_in_air | PM10 mass mixing ratio in lowest model layer | | AERday (daily), AERhr (hourly) |
| sfpm25 | mass_fraction_of_pm2p5_ambient_aerosol_particles_in_air | PM2.5 Mass Mixing Ratio in Lowest Model Layer | existing physical parameter | AERday (daily) |
| so | sea_water_salinity | Sea water salinity | existing physical parameter | Oday (daily) |
| tas | air_temperature | temperature at 2m above surface | existing physical parameter | E1hr (hourly) |
| taUTLS | air_temperature | temperature in the UTLS region | 6 hourly instantaneous temperature in the UTLS region (100, 150 and 225 hPa) | 6hrPlevPt (6-hourly) |
| taUTLSadd | air_temperature | temperature in additional levels within the UTLS region | 6 hourly instantaneous temperature in additional levels within the UTLS region (175 and 225 hPa) | 6hrPlevPt (6-hourly) |
| tauuo | downward_x_stress_at_sea_water_surface | Sea Water Surface Downward X Stress. The stress on the liquid ocean from interactions with overlying atmosphere, sea ice, ice shelf, etc. | existing physical parameter | 3hr (3-hourly) |
| tauvo | downward_y_stress_at_sea_water_surface | Sea Water Surface Downward Y Stress. The stress on the liquid ocean from interactions with overlying atmosphere, sea ice, ice shelf, etc. | existing physical parameter | 3hr (3-hourly) |
| thetao200 | sea_water_potential_temperature | Sea Water Potential Temperature at 200 meters | existing physical parameter | Oday (daily) |
| ts | surface_temperature | Surface temperature (skin for open ocean) | existing physical parameter | E1hr (hourly) |

| | | | | |
|---|---|---|---|---|
| ua100m | eastward_wind | Eastward wind at 100m above the surface | existing physical parameter | E1hr (hourly), 3hr (3-hourly) |
| uas | eastward_wind | Surface wind speed Eastward Components at 10m above the surface | existing physical parameter | E1hr (hourly) |
| uaUTLS | eastward_wind | Eastward wind in the UTLS region. | 6 hourly instantaneous Eastward wind at 5 pressure levels in the UTLS region (150, 175, 200, 225, and 250 hPa) | 6hrPlevPt (6-hourly) |
| uos | surface_sea_water_x_velocity | surface prognostic x-ward velocity component resolved by the model. | | Oday (daily) |
| va100m | northward_wind | Northward wind at 100m above the surface | existing physical parameter | E1hr (hourly), 3hr (3-hourly) |
| vas | northward_wind | Surface wind speed Northward Components at 10m above the surface | existing physical parameter | E1hr (hourly) |
| vaUTLS | northward_wind | Northward wind in the UTLS region | 6 hourly instantaneous Northward wind at 5 pressure levels in the UTLS region (150, 175, 200, 225, and 250 hPa) | 6hrPlevPt (6-hourly) |
| vos | surface_sea_water_y_velocity | surface prognostic y-ward velocity component resolved by the model. | | Oday (daily) |
| wbgt2m | wet_bulb_globe_temperature | mean 2m daily wet bulb globe temperature. Wet Bulb Globe Temperature (WBGT) is a particularly effective indicator of heat stress for active populations such as outdoor workers and athletes. | WBGT = 0.567 * T_C + 0.393 * e/100 + 3.94, where T_C is temperature in degrees C, and e = huss * p * M_air / M_H2O, where "huss=specific humidity in kg/kg", M_H2O = 18.01528/1000 # kg/mol, M_air = 28.964/1000 # kg/mol for dry air and "P = surface pressure in Pa" | day (daily) |
| wbgt2mmax | wet_bulb_globe_temperature | maximum 2m daily wet bulb globe temperature. Wet Bulb Globe Temperature (WBGT) is a particularly effective indicator of heat stress for active populations such as outdoor workers and athletes. | WBGT = 0.567 * T_C + 0.393 * e/100 + 3.94, where T_C is temperature in degrees C, and e = huss * p * M_air / M_H2O, where "huss=specific humidity in kg/kg", M_H2O = 18.01528/1000 # kg/mol, M_air = 28.964/1000 # kg/mol for dry air and "P = surface pressure in Pa" | day (daily) |
| wetnh3 | minus_tendency_of_atmosphere_ mass_content_of_ammonium_dry _aerosol_particles_due_to_wet_d eposition | Wet Deposition Rate of NH4 | existing physical parameter | AERday (daily) |
| wetnh4 | minus_tendency_of_atmosphere_ mass_content_of_ammonium_dry _aerosol_particles_due_to_wet_d eposition | Wet Deposition Rate of NH4 | existing physical parameter | AERday (daily) |
| wetnoy | minus_tendency_of_atmosphere_ mass_content_of_noy_expressed | Wet Deposition Rate of NOy | existing physical parameter | AERday (daily) |

| | | | | |
|---|---|---|---|---|
| | _as_nitrogen_due_to_wet_deposition | | | |
| wfo | water_flux_into_sea_water | Water Flux into Sea Water | existing physical parameter | 3hr (3-hourly) |
| wo | upward_sea_water_velocity | Sea Water Vertical Velocity | existing physical parameter | Oday (daily) |
| wsgmax10m | wind_speed_of_gust | Maximum Speed of Wind Gust at 10m | existing physical parameter | E1hr (hourly), Emon (monthly) |
| wsgmax100m | wind_speed_of_gust | Maximum Wind Speed of Gust at 100m | existing physical parameter | E1hr (hourly), Emon (monthly) |
| zg | geopotential_height | Geopotential height | existing physical parameter | 6hrLev (6-hourly) |
| zmeso | mole_concentration_of_mesozooplankton_expressed_as_carbon_in_sea_water | Mole Concentration of Mesozooplankton Expressed as Carbon in Sea Water | existing physical parameter | Oday (daily) |
| zmicro | mole_concentration_of_microzooplankton_expressed_as_carbon_in_sea_water | Mole Concentration of microzooplankton Expressed as Carbon in Sea Water | existing physical parameter | Oday (daily) |
| zooc | mole_concentration_of_zooplankton_expressed_as_carbon_in_sea_water | Zooplankton Carbon Concentration | existing physical parameter | Oday (daily) |
| zos | sea_surface_height_above_geoid | Sea Surface Height above Geoid | existing physical parameter | Oday (daily) |

**Note: Variables with existing physical parameters were requested at higher temporal resolution, including existing monthly variables requested at daily

resolution, and existing daily resolution variables requested at sub-daily resolution (as fine as 1-hourly).

912

## Code and data availability

The variables and their metadata included latest CMIP7 Assessment Fast Track Data Request can be accessed via the Data Request Task Team. At the time of this publication, the latest major release (v1.2) is accessible at Zenodo under https://doi.org/10.5281/zenodo.15116894 (Data Request Task Team, 2025a), and the latest minor release (v1.2.1) is accessible at Zenodo under https:/doi.org/10.5281/zenodo.15288187 (Data Request Task Team, 2025b).

## Author contributions

ACR prepared the manuscript with contributions from all co-authors. ACR, MJ, BT and CLP provided conceptualization with inputs from CMIP Team. ACR and BT developed the outreach and input methods with help from the CMIP Team. CLP and MJ developed the methods for data inputs and oversaw opportunity evaluation as part of their roles on the CMIP Data Request team. ACR, CT, DJB, CB, ID, JF, PLMG, BH, VH, UI, DI, ILL, TL, XL, JM, NN, SP, IR, W-LT, AW, LZ, and DZ conducted outreach to I&A communities and then processed and summarized their inputs into original manuscript draft preparation. Data curation contributions were provided by ACR, CLP, DJB, JF, PLMG,VH, and BT and ACR and BT provided contributions to visualization. Review and editing was carried out by ACR, BT, CB, ID, DJB, VH, DI, ILL, IR, TL. XL, JM, NN, AW and DZ. BT provided resources and project administration support.

## Competing interests

Author ACR is Co-chair of the VIACS Advisory Board and an employee of NASA.

Author BT is an employee of HE Space Ltd which delivers the CMIP IPO service to the European Space Agency and is Vice Chair of the Chartered Institution of Building Services Engineers Knowledge Generation Panel

Author CT is Co-chair of the VIACS Advisory Board

Author CB is the director of the Copernicus Climate Change service at ECMWF

Author PLMG is employed by the Met Office, UK

Author VH is employed by CSIRO

Author SP is employed by EDF Lab, lab of an electricity company

Author AW is employed by Riskthinking.AI

Our authors making these declarations do so in accordance with the Competing Interests Policy but do not believe their positions have influenced their ability to provide complete and objective presentation of this research.

## Acknowledgments

The Impacts and Adaptation Theme Author Team acknowledges the valuable contributions from a widespread scientific community who participated in the data request effort and public consulting processes including members of the ESM groups who participated in drop-in and feedback webinars. Members of the CMIP7 Vulnerability, Impacts, Adaptation and Climate Services (VIACS) Advisory Board helped develop initial ideas for the data request and aided in efforts to request community inputs. We thank Eleanor O'Rourke and Gang Tang for their helpful comments on the draft. We thank Natalie Kozlowski at Columbia University for design contributions to Figure 1 and Elisabeth Dingley for her conceptualization and design of Figure 2. PLMG acknowledges the contributions of Dr Erika Palin for supporting the interactions with the transport sector, and Dr Paul Williams for his support of the aviation applications opportunity. The author team acknowledges all authors of the opportunities that underpin this paper, and appreciates the suggestions from two anonymous reviewers. The manuscript also benefited from a helpful health applications community comment from Kris Ebi and a comment with additional insights on energy applications from Rémi Gandoin, Andrea Hahmann and Jan Wohland.

## Financial Support

ACR contributions were supported by a NASA Community Service Grant and Earth Science Division support of the NASA GISS Climate Impacts Group. The work of CLP was supported by the National Centre for Atmospheric Science (NCAS, UK). UI acknowledges European Union's Horizon Europe Programme under CleanCloud (No. 101137639). JM was supported by the Reducing Uncertainties in Biogeochemical Interactions through Synthesis and Computing Scientific Focus Area (RUBISCO SFA) project funded through the Earth and Environmental Systems Sciences Division of the Biological and Environmental Research Office in the US Department of Energy (DOE) Office of Science. Oak Ridge National Laboratory (ORNL) is supported by the Office of Science of the DOE under Contract No. DE-AC05-00OR22725. DZ contributions were supported by the German Science Foundation DFG (Grant no. ZU 361/6-1). The work of DI was supported by the Foundation Euro-Mediterranean Center on Climate Change (CMCC, Italy). BH received funding from the European Union's Horizon 2020 Research and Innovation program under grant agreement no. 101003536 (ESM2025 – Earth System Models for the Future). The work of CB was supported by the Copernicus Climate Change Service, a programme of the European Commission implemented by ECMWF. The work of MJ was supported by UK Research and Innovation (grant no. NE/Y001729/1). The work of XL was supported by U.S. NSF #2420405. The work of LZ was supported by US NSF CAREER Award no. 2145362; US NASA no. 80NSSC25K7322. BT is a staff member of the CMIP IPO which is hosted by the European Space Agency, with staff provided on contract by HE Space Operations Ltd. Views and opinions expressed are however those of the author(s) only and do not necessarily reflect those of their employers or funders. Neither the European Union nor the granting authority can be held responsible for them.

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
