# Peer review of "CMIP7 Data Request: Impacts and Adaptation Priorities and 1"

_EGUsphere, 2025_

## Community Comment (CC2)

**Comment of "CMIP7 Data Request: Impacts and Adaptation Priorities and Opportunities" by Ruane et al., 2025.**

Dear authors,

Thank you for a clear and well-written article, providing in-depth information on the CMIP7 variable selection process.

The purpose of this community comment is to bring your attention to a topic of our concern, namely dynamic changes in land use in Earth System Models (ESM) and their impact on boundary layer winds and other physical variables relevant for Wind Energy and Renewables. In short:

- We are grateful that the CMIP7 dataset will include the 100m wind, as it is not trivial to compute from standard output and the effect of surface roughness is expected to be smaller than for the 10m wind. This reduces uncertainties in our work.
- However, additional information may be required for the users to manage the uncertainty associated with land use change, and we propose some suggestions for downstream applications of CMIP7 dataset which some of the authors of this paper may find relevant and actionable (within the Copernicus Climate Change Service for instance).

This comment was initiated by Rémi Gandoin (C2Wind, Denmark) and subsequently reviewed and amended by Andrea Hahmann (DTU, Denmark) and Jan Wohland (University of Oslo, Norway). They form a small group of practitioners and researchers primarily, all having published studies on the effect of Climate Change on Wind Energy, see the journal articles (Hahmann et al., 2022) (Wohland, 2022) (Wohland et al., 2024) and the technical publications (C2Wind, 2024) (C2Wind, 2025).

**Problem statement**

All other things being equal, changes in land use can lead to noticeable changes in boundary layer winds, and thereby it can be difficult to quantify in wind resources between future and historical scenarios when using ESM model where land use changes dynamically (Wohland et al., 2024) (Collet et al., 2025).

We do of course acknowledge that surface roughness changes over time and that this needs to be accounted for, however it is important for analysts and scientists to be able to quantify how much of the change in wind resource is due to changes in roughness, and how much is due to

changes in global/regional weather patterns. Some of the land-use changes in the SSP scenarios are highly idealised like the ones in Poland, and they have a strong impact on climate change impact assessments for wind energy in these areas.

We acknowledge that, assuming perfect and seamless access to the entirety of the datasets (including model levels) described in your article, users may be able to perform the required analysis to untangle land use and global/regional circulation effects (using data higher up in the troposphere helps decrease the effect of land use even further). However, based on our experience with CMIP6 datasets, we foresee that an infinitesimally small fraction of industry users will make use of the ESM native or pressure level due to the difficulty in accessing these data. As of today:

> Most practitioners only use single level data available via third party services such as the Copernicus Data Store (CDS).
> Advanced users (typically in academia) can only "afford" (time wise) to use only a subset of ESM datasets at pressure- or native model levels.

**Remedial measures**

As the CMIP7 modelling phase is about to start, we would like to argue about initiating the actions suggested below. We acknowledge that these fall outside the original scope of your work, but we take the opportunity to present them here to all co-authors of this article with the hope that some of them could be activated as part of downstream application projects such as the CDS here taken as an example.

1) Provide, via a platform like the CDS, pressure levels wind time series with at least daily resolution. To our knowledge, only monthly time series are available from the CDS.

2) Provide, via a platform like the CDS, ESM time series related to the land use type and composition. These could include for instance the below existing variables, but we seek your feedback whether additional variables may be useful too (for instance, we could not find variable describing urban land use types). Note: the names below are the ones reported in the CMIP7 variable database provided with the article. We are unsure whether/how "height" translate to "aerodynamic roughness length", see the next item.
   - Height of Crops
   - Height of Grass
   - Height of Pastures
   - Height of Shrubs
   - Height of the Vegetation Canopy

- Height of the Vegetation Canopy
- Height of Trees.

3) Provide synthetic information (in the form of table with references to sections of ESM documentation) regarding:
   - How land use is accounted for in the surface layer module, for instance see https://escomp.github.io/CTSM/release-clm5.0/tech_note/Ecosystem/CLM50_Tech_Note_Ecosystem.html.
   - Furthermore, the height of the various vegetation types is not always how the surface roughness length is calculated. Each LSM does it differently. The surface aerodynamic roughness length is a more "physical" variable that every land surface modeler and can be directly related to shear and simulated wind profiles; obtaining these values (if they exist, would be very useful).
   - How the single level winds are computed (including information on interpolation method, or any diagnostic procedure).

Could these suggestions be brought forward in future discussions between CMIP7 and other parties concerned with downstream applications such as the C3S?

**Additional request**

Would it be possible to provide a single csv/yaml file with all CMIP6/7 variable descriptions contained in the json files attached to the article? We have created a test version in all_vars.xlsx, which we believe could be useful to the users not having the time to process all of the json information.

**References**

Collet F., Boé J., Bador M., et al. Future forest cover expansion drives a decrease in near-surface wind speed in Europe in global climate models. *ESS Open Archive* . June 25, 2025.

Davis, N. N., and Coauthors, 2023: The Global Wind Atlas: A High-Resolution Dataset of Climatologies and Associated Web-Based Application. *Bull. Amer. Meteor. Soc.*, **104**, E1507–E1525, https://doi.org/10.1175/BAMS-D-21-0075.1.

Denager, T., Sonnenborg, T. O., Looms, M. C., Bogena, H., and Jensen, K. H.: Point-scale multi-objective calibration of the Community Land Model (version 5.0) using in situ observations of water and energy fluxes and variables, Hydrol. Earth Syst. Sci., 27, 2827–2845, https://doi.org/10.5194/hess-27-2827-2023, 2023.

Hahmann, A. N., García-Santiago, O., and Peña, A.: Current and future wind energy resources in the North Sea according to CMIP6, Wind Energ. Sci., 7, 2373–2391, https://doi.org/10.5194/wes-7-2373-2022, 2022.

Wohland, J. Process-based climate change assessment for European winds using EURO-CORDEX and global models. Environmental Research Letters. https://doi.org/10.5194/esd-15-1385-2024, 2022.

Wohland, J., Hoffmann, P., Lima, D. C. A., Breil, M., Asselin, O., and Rechid, D.: Extrapolation is not enough: impacts of extreme land use change on wind profiles and wind energy according to regional climate models, Earth Syst. Dynam., 15, 1385–1400, https://10.1088/1748-9326/aca77f, 2022

C2Wind. Quantitative approach for climate model selection and application in WRA & EYA, https://c2wind.com/f/content/pres08ge_v2.pdf, 2024.

C2Wind. Quantifying and Validating the Impact of Climate Change on Wind Resources in the Mediterranean Sea and Southern North Sea, https://c2wind.com/f/content/po087_v3.pdf, 2025.

---

## Author Response (AR1)

Author Responses to Reviewer Comments for:

**CMIP7 Data Request: Impacts and Adaptation Priorities and Opportunities**

*Ruane et al., Submitted July 15[th], 2025, to GMD*

*Updated: October 9th, 2025*

**Reviewer RC1:**

This paper outlines the process that was used for gathering and refining the data request for CMIP7.  For this particular data request version, they surveyed a broader community, including the communities of impacts and adaptation applications that use Earth System Model outputs.  This was done to purposely address the disconnect between the ESM's output and the diverse needs of the impacts community.  This is an important consideration and will allow easier use of the data in downstream applications that are used for risk mitigation and adaptation planning and will foster better collaboration and engagement between different communities.

The work described in this paper is essential to communicating the data requirements of this large global effort.  And I applaud the authors for considering a larger audience in the use of these important datasets.

Authors' Response: We thank the Reviewer who provided RC1 for their consideration of the submission and for making suggestions that have improved the quality of the revised manuscript.  We have answered each review comment element below.

There were a couple of sections in this document where it would have been helpful to have some more information.  These included:

1. pg 6, section 2: I would like to see a little bit more detail on the number of different groups/communities that supplied information into the request and if the impacts community was equally represented on the I & A team (was there equal input into the different Opportunities/Impacts/Variable Groups)?  Did you consider the engagement/outreach successful?

   Authors' Response: We have added an additional short paragraph at the end of Section 2 to reflect on the success of the process as it played out. This allows further information about the types of groups most likely to respond, the breadth of the inputs, the way we filled gaps and the way we tracked the iterative process in determining variable group priorities for each opportunity.  We also note the likelihood that additional groups did not explicitly engage in the process.

2. pg 9: While it was indicated that not all Variable groups are mandatory and some are higher priority, it wasn't clear if it's required to supply all variables within a Variable group.  Could extra clarity be added here?

Authors' Response: At the end of section 4.1 there is a discussion on prioritization given resource constraints.  There, we note that even a subset of variables from a variable group would be valuable: "The I&A Team also encourages ESM groups to avoid thinking that opportunities are all-or-nothing; even providing variables for one Variable Group (or a subset of those variables) is valuable."

Suggestions for technical corrections:

1. Section 4.1.4 was difficult to read, with many sentences having extra information in parentheses.  Could this be rewritten to flow more smoothly and match the style of the other similar sections around it?

Authors' Response: We have revised Section 4.1.4 to improve readability. Specifically, we reduced parentheses by integrating explanations, simplified sentences, and clarified the distinction between the "core" and "expanded" variable groups.

2. Line 31, pg 13: The parentheses before Craig should be removed so the number of parentheses matches.  This might be addressed in #1, though.

Authors' Response: We understand the reviewer to have meant line 315, where we have corrected the open/close parentheses around the entire e.g. statement that included a reference to Craig et al.

3. Line 360 talks about an idea that was talked about on pg 9.  Not sure if it needs to be repeated here.  "ESM groups supporting opportunity are not mandated to provide all the requested variable groups. Contributors can focus on the core variable groups and are encouraged to provide as many of the additional groups as they can afford to support the emerging components in regional Earth system modeling"

Authors' Response: We have revised the end of Section 4.1.5 to note that the large size of the Dynamical Downscaling opportunity necessitates reiterating the need for practical resource use, beginning with the core datasets. This avoids redundancy while emphasizing a case where ESMs are likely to run into resource constraints – the I&A Team feels it is important to convey that contributors should not be frightened away by the overall opportunity size when the core groups may be more feasible.

4. Line 413, pg 17: This paragraph again uses several parentheses and is more difficult to read, and doesn't match the writing style of the other sections around it.  Could it be rewritten to flow better?

Authors' Response: We have combined successive parenthetical statements for clarity.

5. In sections 4.1.12, 4.1.13, 4.1.14, opportunity ID55, opportunity ID 49:  Tables 13, 14, 16, and 17 don't seem to be referenced in the text.  Is this on purpose for style or was this an oversight?

Authors' Response: This was an oversight. We now reference Tables 13, 14, 16, and 17 in the text, resulting in a more cohesive presentation of text and tables.

6. Troposphere is mispelled as "trposphere" on pg 37 (Table row: ID 41)

Authors' Response: Thanks for catching this – we have corrected to 'troposphere' in Table A.1.

**Reviewer RC2:**

This is a well-written and well-organized paper. The authors have clearly invested significant effort in preparing a comprehensive and structured study. While some of the recommendations may be challenging for Earth System Model (ESM) groups to execute in full, the work provides an important reference point. It can serve as a target for groups aiming to save additional fields or guide future model development in cases where certain processes are not yet represented in the models. One particularly valuable addition would be the use of metrics (e.g., from CMIP6) to show which Impact & Adaptation (I&A) variables are most widely used. Since ESM groups are often constrained by resources and cannot always write out large, high-resolution datasets, such metrics would help prioritize outputs and make the recommendations more practical. Overall, the manuscript represents a valuable contribution to the community and requires only minor clarification and technical corrections before publication.

Authors' Response: We thank the Reviewer who provided RC2 for their consideration of the submission and for review comments that point us toward a more clear and impactful manuscript.  We have answered each review comment element below.

Regarding metrics of CMIP6 variables, the download counts for CMIP6 variables were calculated by the Data Request Team, but we elected not to focus heavily on this metric in designing CMIP7 for several reasons. First, our aim was to solicit organic expressions of interest from I&A communities and need rather than skewing with pre-populated metrics that could drive users toward specific incumbent variables. Second, download counts reflect interest in 'best available' outputs, not necessarily 'most appropriate', 'most desired', or 'most applicable' variables; each of which are of greater utility for this I&A Data Request. Along the same lines, ESMs and impact and adaptation tools have changed since CMIP6 was released and this would affect interest in variables. Third, as noted in the introduction, we have an interest in identifying

variable groups that are tailored to the needs of communities that may not have a high download count even as they are contributing important applications – these groups would be overlooked in a download count prioritization.

Minor comment and technical issues:

1. Line 155: please briefly discuss Mural Board for readers who are not familiar.

Authors' Response: We have added an additional sentence describing the Mural Board web application and the way that respondents indicated variables and provided justifications for variable sets.

2. Figure 2 has a nice chart to show community engagement, somehow it is hard to parser, especially for the outer most ring the different some color of the user groups are not labeled.

Authors' Response: We appreciate the reviewers' feedback on Figure 2, which will improve the presentation in this paper as well as in other CMIP7 Data Request papers that also include a similar figure colored according to each Data Request group's outreach. This allows for cohesive communication and comparability between data request papers.  As each manuscript is moving through the review process at a slightly different pace, we anticipate that another update to improve clarity will be available before final publication. For this version, we added additional clarifications to the caption to indicate that the colors indicated in the bottom-right legend are for the inner circle broader categorizations, with those same categories fanning out to the outer ring where outreach to sub-groups is specifically assessed.

3. Line 248: please define ISIMIP when it first appear (which was defined later in the table)

Authors' Response: ISIMIP is first defined in Section 4.1.1 where we spell out the full acronym.

4. Section 4.1.6: please provide brief discussion on empirical statistical downscaling (ESD) and emulator, because later traditional ESD and hybrid emulator are mentioned but is not explained.

Authors' Response: We have provided an enhanced explanation of empirical statistical downscaling (ESD) when it first appears in the text: "Empirical-statistical downscaling (ESD; Maraun et al., 2019) exploits the relationship between large-scale and local climate observations to train models that convert coarse ESM output into regional-to-local climate projections. Mathematical models can also be trained on complex ESM and RCM outputs to reproduce key features at lower computational cost, thus statistically emulating its behaviour (Baño-Medina et al., 2024)."

5. Line 633: it mentioned 93 variables here, but the number of variables was 539 unique variables (on line 49). Is it a typo?

Authors' Response: We have clarified this line to note that these are 93 new variables (not previously included in CMIP): "These I&A opportunities and variable groups included 539 unique variables, including 93 new variables defined for CMIP7 in Appendix B)."

6 Line 591-592: missing subject in the sentence.

Authors' Response: We have clarified this sentence to now include the Rapid Evaluation Framework (REF) as the subject: "REF was created to evaluate and benchmark the newly available CMIP7 AFT simulations as soon as they are uploaded to the Earth System Grid Federation (ESGF) with metrics and diagnostics that are available through different open-source evaluation and benchmarking tools (Hoffman et al. 2025)."

7. Line 858: "...clear documentation of outputs is needed..."

Authors' Response: We have clarified the grammar in this sentence, which now matches the recommended phrasing.

8. Appendix A. Opportunity processing: consider to revise the title of Appendix A to be more descriptive.

Authors' Response: We have revised the title of Appendix A to read: "Data Request Opportunity Evaluation and Processing"

**Reviewer CC1 (Kris Ebi):**

The authors are to be congratulated on a valuable and inclusive process for obtaining data request opportunities from the impacts and adaptation community.

Authors' Response: Thanks, Kris, for your deeper examination of the manuscript. Your comments will help us better reflect the variable needs and opportunities for the health sector, as well as to more effectively drive readers to variables which may aid in health assessments even as they are more closely associated with other systems interconnected with health. We have addressed your suggestions below.

Two suggestions for accuracy and clarity.

Section 4.1.8: human health. This section does not describe the full health risks of a changing climate. The "indirect" impacts of climate change on health are much more than air pollution and include a wide range of vector borne diseases, food and water safety and security, malnutrition, and the consequences of migration and displacement. It would be helpful to understand the level of engagement across the health sector. This means that Table 9 does not reflect the range of variables used in health projections.

Authors' Response: We have deepened the discussion around health risks in Section 4.1.8 to better reflect the breadth of hazards and risks, including those through systems that are

components of the broader health systems (e.g., food, water, ecosystems, urban areas), for example: "Climate change has both direct and indirect impacts on human health. Direct health impacts arise from a number of climatic impact-drivers, including pervasive mean conditions and extreme heat, cold, wind storm, flooding, and ice events, as well as changes to atmospheric composition which influence air pollution levels humans are exposed to (Im et al., 2022, 2023). Indirect health risks arise from climate's influence on the many natural and human systems that are linked to broader health systems, including effects on vector-borne diseases, food and water safety and security, malnutrition, and the consequences of malnutrition and displacement."

We also note that the interconnectedness of the health sector merits attention to cross-cutting variables requested in other opportunities (such as those needed for water quality or food insecurity) found in water sector or food data requests, should the reader be interested in pursuing those further. This is reflected in both the text and Table caption, for example: "Additional variables relevant to indirect health CIDs are available in other I&A opportunities related to food systems (ID 19), water systems (ID 67), ecosystems (IDs 11 and 16), urban populations (ID 40), and broader climate services (ID 20)."

Line 793 includes the recommendation for sharing best practices, tutorials, and guidelines. Do the authors have a recommendation for how that could be accomplished?

Authors' Response: We prefer not to be too prescriptive in making specific suggestions in this realm, as we are hoping to motivate the reader to consider ways that best practices, tutorials, and guidelines could be developed and shared. As a specific way for CMIP to help move this forward (and a potential starting point for reviewers, we note the importance of continuing collaboration with relevant major programs in this area: "For example, we encourage continuing collaborations around CMIP7 data provision, access, further processing and applications between CMIP and the World Climate Research Programme's Regional Information for Society (RIfS) Joint Task Team on Responsible Data Use, as well as with the World Adaptation Science Program (WASP)."

**Reviewer CC2 (Rémi Gandoin and Team):**

Review Comment CC2, which includes well-supported recommendations for further development of wind energy variables, model documentation, variables relating to land use changes and surface roughness, variable documentation, and data distribution, here: https://doi.org/10.5194/egusphere-2025-3408-CC2

Authors' Response: We thank the reviewer group CC2 (Gandolin, Wohland, Hahmann) for their detailed comments. We start our response by noting, however, it is beyond the scope of this paper and the authors' remit as participants in the CMIP data request team to propose additional "Opportunities" or "Variable Groups" at this stage. We will take this information into consideration as future versions of the Data Request are considered, as the CMIP process will continue to be responsive to feedback even as frameworks and decisions are being made to keep all participants on schedule. Also, although daily model level wind time series are not specifically requested by the Impacts & Adaptation group, these variables may become available elsewhere from within the wider I&A Data Request (e.g., 6-hourly 3D data are requested as part of the Dynamical Downscaling Opportunity ID 81) and broader CMIP Data Request. We would therefore encourage the reviewers to familiarise themselves with what has been requested more widely via the CMIP AirTable.

In terms of the specific proposal for daily/sub-daily wind-vectors on model levels, we note that this opportunity would likely be low priority given that the reviewers themselves suggest that the variables may have limited uptake (as few users will have the skills / know-how and capacity to process them). Moreover, the data volumes involved in model-level vector wind-fields are non trivial. These concerns would not necessarily prevent their inclusion in a data request but, in both aspects, there is a tension with the overall need to support a wide range of users and for the total data volume to be minimized. We hope that this illustrates the trade-offs that need to be considered without necessarily prejudging which is the 'best use' of CMIP modelling group's efforts.

Regarding the "remedial actions" proposed by the commenters:

1. Availability of specific CMIP output via specific platforms (such as EU Copernicus CDS) is beyond the scope of this paper and the CMIP7 Data Request Team activities. We do, however, agree that as much data as possible should be made accessible as quickly as possible via the relevant CMIP distribution nodes, and have provided this input to the CMIP7 planning process on numerous occasions.

2. As noted above, it is beyond the scope of this paper to request additional variables or to make changes to the data request at this stage. We agree, however, that it would have been good to have included key land-surface information as part of the energy systems request. Nevertheless, many of these variables are potentially either model inputs (e.g., models lacking dynamic land surface models) or may be requested as outputs elsewhere in the CMIP data requests (e.g., the Agriculture and Food Systems Impacts Opportunity ID 19 requests information about agricultural land use, and relevant information may also be provided by ScenarioMIP and LUMIP). Again, we encourage the reviewers to familiarise themselves with the CMIP AirTable for a full view of the data request if there are specific variables they wish to explore.

3. We agree with the reviewers that documentation of model design and diagnostic calculations are important issues. Typically, this information is recorded in individual model documentation papers, though we appreciate this may not necessarily be complete or as accessible as one might wish. We anticipate that such documentation may also appear in studies of model diagnostics as part of the DECK simulations analysis. We note, however, that this issue goes well beyond the scope of this paper to address.

Regarding the final 'additional request' for CMIP variable descriptions as a CSV/YAML files: we agree this could be useful though, again, this goes beyond the scope of this particular piece of work. We note, however, that the AirTable is already a user-friendly way to explore the data request in tabular format. We also understand additional ways to view the latest version are underway, and encourage continuing engagement with the CMIP7 International Program Office about such ideas.